# Local Network Interaction as a Mechanism for Wealth Inequality

Shao-Tzu Yu [1,2] ✉, Peng Wang [3], Chodziwadziwa W. Kabudula[4], Dickman Gareta [5], Guy Harling [4,5,6,7] & Brian Houle [2,4,8]

Given limited institutional resources, low-income populations often rely on social networks to improve their socioeconomic outcomes. However, it remains in question whether small-scale social interactions could affect large-scale economic inequalities in under-resourced contexts. Here, we leverage population-level data from one of the poorest South African settings to construct a large-scale, geographically defined, inter-household social network. Using a multilevel network model, we show that having social ties in close geographic proximity is associated with stable household asset conditions, while geographically distant ties correlate to changes in asset allocation. Notably, we find that localised network interactions are associated with an increase in wealth inequality at the regional level, demonstrating how macro-level inequality may arise from micro-level social processes. Our findings highlight the importance of understanding complex social connections underpinning inter-household resource dynamics, and raise the potential of large-scale social assistance programs to reduce disparities in resource-ownership by accounting for local social constraints.

The ability of people to draw upon support from social networks is an essential strategy for coping with social and financial adversity. Theoretical work suggests that these connections (ties) between individuals (nodes) serve as channels for accessing valuable information, resources, and supports, thereby having a major influence on the economic development of human society[1–5]. This line of research highlights that inequalities may arise socio-economically when resources flow through a social network structured by geography[6,7] or socio-demographic characteristics[8–11]. Hence, policy efforts that aim to harness social connectivity between people with diverse social backgrounds have been suggested to induce equal opportunities[12]. Despite the potential of social network research to elucidate disparities in economic opportunities, however, its scientific progress is currently impeded by three distinct theoretical and empirical challenges.

First, limited studies have been conducted in low-income populations[2,4]. Previous research has highlighted the importance of social networks as critical support systems in poorer environments[13–15]. Yet, empirical investigation exploring the influence of social networks on inequalities in these settings pose a significant scalability challenge. This limitation may be hampered by high costs and inadequate infrastructure for extensive research activities, particularly among those located in remote settings. As a result, existing research tends to focus on specific population sub-groups[16], lacking data on network interactions observed at the population level. In response, computational and online experimental techniques may offer scalable alternatives[7,9,11]. However, it remains in question how these digital proxies can capture more tangible, costly, and culturally defined social interactions related to resource-sharing and exchange activities among families in poverty-

¹Office of Population Research, Princeton University, Princeton, NJ, USA. ²School of Demography, The Australian National University, Canberra, ACT, Australia. ³Centre for Transformative Innovation, Swinburne University of Technology, Melbourne, Australia. ⁴MRC/Wits Rural Public Health and Health Transitions Research Unit (Agincourt), School of Public Health, Faulty of Health Science, University of the Witwatersrand, Johannesburg, South Africa. ⁵Africa Health Research Institute, Durban, South Africa. ⁶Institute for Global Health, University College London, London, UK. ⁷School of Nursing & Public Health, College of Health Sciences, University of KwaZulu-Natal, Durban, South Africa. ⁸CU Population Center, Institute of Behavioral Science, University of Colorado at Boulder, Boulder, CO, USA. ✉e-mail: shaotzuyu@princeton.edu

stricken settings[17,18]. The availability of population-level in-person data therefore offers an important opportunity to gain a deeper understanding of the under-researched correlation between social networks and economic inequality in rural or poorer contexts.

Second, most existing research has overlooked the intricate and multi-layered network structures observed in real-world social networks, including one's affiliation in groups[19], residency in places[19,20], and their connections to ecological spaces[21]. The accelerated urbanisation and globalisation of modern societies have altered the ways in which people are connected[22], particularly in the Global South[23]. This pattern is evident among rural populations with limited job prospects, whose livelihoods rely primarily on economic support from distant sources[24]. These physical and geographic boundaries may have a considerable influence on a range of economic outcomes for the interconnected individuals[22], as well as the prosperity of these communities at large[7,25]. Nevertheless, little is known about the degree to which these network interactions – spanning across diverse geographic boundaries – are associated with the economic development of rural communities. As such, an alternative, multilevel social network framework is required to better estimate the effect of these geographically defined network structures on social inequalities.

Third, several longstanding questions in social and natural sciences relate to the puzzle of whether small-scale social processes can lead to emergent phenomena at the macro level[26–30]. This micro-to-macro question, pertinent in understanding phenomena such as the recent COVID-19 pandemic[31] and climate change responses[32], seeks to clarify how local social interactions might traverse into population-level dynamics. However, understanding these micro-macro linkages remains a formidable challenge, often due to the difficulties of harmonising and modelling the expected social patterns and the inherent randomness in human social interactions[33]. Previous research has explored the association between macro-level inequality and global network properties[7,34], as well as individual-level economic outcomes and local social interactions[35–37]. Yet, the link between macro- and micro-level economic outcomes, alongside the specific types of social interactions that may strengthen returns capable of influencing economic disparities, remains poorly understood. Unpacking the varying network dynamics at play is therefore likely to contribute to a more refined understanding of how to enhance the effectiveness and efficiency of social policy interventions[31,38].

Here, we investigate how, and to what degree, the disparities in economic resources may be accentuated by various forms of micro-level social interactions in one of the poorest rural South African settings that has endured repercussions from decade-long racial segregation (apartheid) and more recently, the HIV epidemic. Our study population is located in the uMkhanyakude district in the KwaZulu-Natal province of South Africa, and comprises the Africa Health Research Institute's (AHRI) Demographic Surveillance Area (DSA)[39,40]. As a predominantly rural economy historically centred around small-scale farming and animal husbandry, our study context is characteristic of many rural South African settings, with limited institutional support resources and employment opportunities[39,40]. Access to healthcare, education, and other amenities is hindered by geographic distance and a lack of public transportation.

To construct social networks on a population scale, our work integrates multiple data sources from AHRI DSA, including a population census, a detailed household residency survey, and information on the geographic locations of households. We leverage a unique census indicator – household memberships – to identify inter-household ties formed by overlapping household members for ~12,000 households (~90,000 people) living across an area of over 400 km². Here, household membership is defined according to one's affiliation to a 'social group' rather than their actual residency. This definition acknowledges the essential role of absent members in the household economy, where shared identity, obligations, or a

common leader foster connections and the flow of resources among social groups[41–43]. Individuals may therefore maintain multiple concurrent household memberships over time. Such conceptualisation has thus challenged the primacy of viewing households as independent units of analysis, especially among rural South African communities[14,43–46].

The significance of these inter-household relationships stems from their role as crucial channels for accessing various forms of resources and support. In lower-income settings, resource pooling primarily occurs informally via social networks given geographic barriers and formal market constraints[17]. These resources can be supplied directly through gifts, transfers, or informal loans. They can also be supplied indirectly by disseminating job opportunities and livelihood strategies such as farming[16]. These networks, predominantly familial, are often bound by culturally defined social contracts, compelling families to voluntarily or involuntarily adhere to sharing, caring, and lending practices[14]. As such, they may also give rise to uneven economic pressures, with benefits for some translating into losses for others[18]. Such reality challenges the presumption that social networks are unequivocally beneficial[47]. To this end, the question of whether these micro-level social processes have the potential to influence the distribution of economic resources for the communities at large remains underexplored.

Here we estimate whether a change in a household's economic conditions is associated with corresponding changes in their network peers. We also examine how these micro-level network interactions can result in observable increases in regional-level inequality outcomes. To do so, we combine the network data with a population-level socio-demographic survey that enables us to measure a household's asset wealth status over time (see 'Household asset wealth' in the Methods section). We then extend this asset-based index to the regional level, summarising the patterns and dynamics of wealth inequality for each of the 23 administrative units within the DSA (singular *isigodi*, plural *izigodi*).

With this multilevel framework in place, our empirical approach rests on the assumption that complex social systems, which encompass social interactions across diverse geographic spaces, can be modelled and represented by a smaller set of locally-specified, multi-level social network configurations comprised of nodes and links embedded in different contexts[48,49]. This approach conceives network ties to emerge, persist, and dissolve through actions of individuals that intersect with those socially proximate others[50]. Under this assumption, the formation of social networks is understood as a self-organising endogenous process, in which node-level effects (household- and regional-level socio-demographics and outcomes) and structural effects (geographically defined network structures) emerge *interdependently* to induce localised social interactions to global responses[28]. This bottom-up approach, therefore, allows us to jointly examine how various smaller-scaled network behaviours are related to the outcome observed at the household level and, more importantly, how they are related to global regional-level inequality patterns.

By employing a multilevel network modelling framework to generalise the economic interdependence among socially connected households on a larger scale, our findings demonstrate that being embedded in diverse network structures, defined by geographic boundaries, reinforces different network effects that are associated with the distribution of economic resources among households. Having proximate social ties is associated with stable household asset conditions, while ties that span greater distances appear to correlate with changes in asset allocation. Notably, the dynamics of economic conditions are more likely to be observed among socially and locally connected households. These local network interactions are associated with an increase in wealth inequality at the regional level, highlighting how inequality may be related to micro-level social processes.

## Results

### Inter-household social network data

Our study population comprises 11,834 households (92,688 people) observed in 2016, scattered across 23 *izigodi*[39]. Demographically, the population is characterised by a high proportion of younger- and middle-aged individuals, with median ages of approximately 22 years for men and 25 years for women in 2018[39]. Non-residents comprise about 28% of the population, of whom are often considered as circular labour migrants engaging in temporary movements seeking support, education, and economic opportunities elsewhere, while remaining socio-economically tied to their rural home[39]. However, the unemployment rate remains steadily high, with nearly 62% of adults lacking formal employment. Consequently, the population relies heavily on non-contributory social grants for household livelihood, particularly the older person's grant, while continuing to be highly mobile[39,41].

To construct the social network for this population, we integrate a prospective, detailed household residency survey spanning from 2000 to 2016 to identify inter-household ties based on overlapping household members. Here, we consider two households as connected if they have shared a common member in the study timeframe (Fig. 1a). AHRI DSA distinguishes one's household memberships (household) with their actual residency status (homestead) to capture the 'stretched households' arrangements observed across many rural South African settings, whereby families work together socio-economically while living apart[43]. In this population, a concurrent membership reflects inter-household ties that are predominantly driven by a combination of shared responsibilities (care receiving and giving), authority (headship and polygamy), and identity (kinship) between households, as well as by historical social relations that may give rise to a sense of belonginess between families[41,43,45]. In these instances, household members are thus usually, but not always, related.

Descriptively, this sparsely connected undirected network consists of 10,162 inter-household ties and follows a long-tailed degree distribution with an average degree of 1.72 (SD = 1.8) per household

(Fig. 1b). This network has a global clustering coefficient of approximately 0.21 and an average path length of roughly 11.84 (SD = 2.98). Over 80% of households, approximately 8453 in total, are connected through a shared member. The largest connected component in this network consists of about 6462 households (Fig. 1c).

We compare this observed network with theoretical models within the broader landscape of network topologies, such as the random[51], small-world[30], and scale-free[26] types (Supplementary Table 2 and Supplementary Fig. 2). Compared to a random network, which has a lower clustering coefficient and a shorter average path length, our network shows a degree of clustering indicative of a more structured, possibly hierarchical arrangement. Although the average path length and local clustering are closer to that observed in small-world networks, the degree distribution does not align with the uniformity seen in such models, nor does it fit the hub-and-spoke configuration typical of scale-free networks. These observations suggest that our observed network may exhibit a community structure with a quasi-small-world configuration, where a small share of social ties could play a role in enhancing overall connectivity and functionality[30].

To further explore the configurations of the constructed network, we delineate the local substructures of the largest connected component using random walks (Fig. 1c). In addition to the typical spanning-tree-like structure found in family lineage networks[45] (Random walk 3), these substructures uncover diverse embeddedness patterns that have not been widely reported in studies concerning the utility and economic functions of informal family support systems across lower-income settings[13–17]. We discuss the setting, data conceptualisation, and network construction method in greater detail in the Methods section and Supplementary Note 1.

### Baseline statistics

We measure economic outcomes at both the household- and regional levels. These outcome variables are derived from the household socio-demographic surveys in 2016 and 2018 (see 'Household asset wealth' in

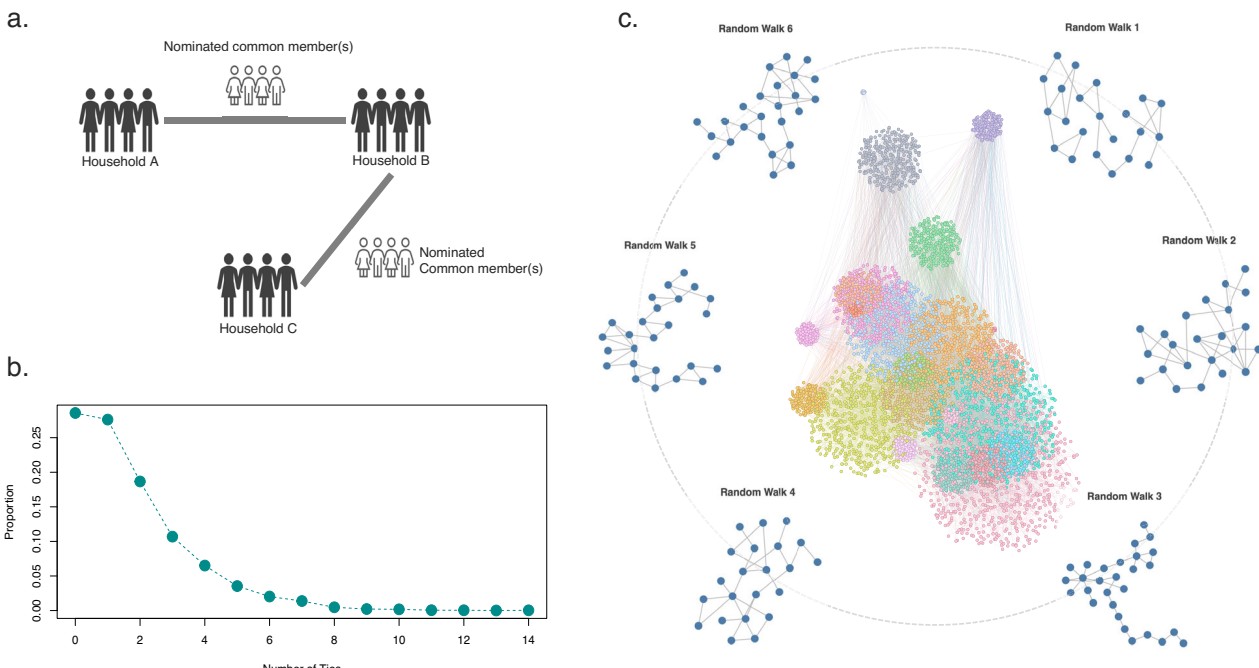

**Fig. 1 | Inter-household social networks.** Panel **a** displays the conceptualisation behind the network data construction method. We considered two households as connected if they have nominated a common household member in our study timeframe, 2000-2016. Panel **b** shows the degree distribution of the inter-household social network. Panel **c** shows the largest connected component, placed and coloured by regions (*izigodi*). To visually elucidate the substructure of the inter-household social network, we conducted six random walks, each comprising 50 steps, within the largest connected component. Random walks were initiated from randomly selected households.

Methods). The data provides a household wealth index that summarises a list of asset items ranging from having durable goods and livestock to power and water supplies. This asset-based index, constructed through Principal Component Analysis (PCA), is commonly divided into five quintiles, ranging from the poorest to the wealthiest households[39].

At the household level (denoted as $h$), we create a binary outcome variable to measure if there has been any change in a household's relative wealth quintile between the two timeframes ($\mathbf{1}_{\{\Delta w_h \neq 0\}}$ where $\Delta w_h = w_{h,2018} - w_{h,2016}$). To understand the disparity in asset wealth at the regional level, we utilise the household wealth index to construct a measure of wealth inequality for each *isigodi* (denoted as $v$), following a similar approach to constructing a Gini index (Supplementary Note 2). We then create a binary variable to identify regions that have experienced an increase in their inequality score over time ($\mathbf{1}_{\{\Delta Inq_v > 0\}}$ where $\Delta Inq_v = Inq_{v,2018} - Inq_{v,2016}$). The household-level binary classification is chosen to identify changes in wealth within households rather than the direction of these changes, since an increase in regional wealth inequality could result from various combinations of upward, downward, and stable asset movements. Given the constraints of our empirical framework (see 'Statistical analyses' in Methods), we conduct three sensitivity analyses where the binary outcome is defined as: i) upward; ii) downward; and iii) no change in asset quintile (see Supplementary Note 3).

Our analysis begins by estimating two baseline logistic regressions predicting outcomes observed at both levels, without accounting for network parameters. We first introduce a range of covariates for both models (see 'Statistical analysis' in Methods). These baseline results demonstrate that over 50% of households experienced a change in their asset wealth status over time (Fig. 2a). The sex of the household head (an indication of an absent male head due to circular labour migration, polygamy, or AIDS-related mortality[45,51]), being eligible for accessing institutional resources such as the old age grant (a primary source of stable income for rural populations[52]), and baseline asset status have a significant influence on subsequent household asset wealth (Fig. 2b).

At the regional level, the overall level of wealth inequality remained relatively stable across the observed timeframe, with an average inequality score of 0.26 in 2016 to an average of 0.25 in 2018. Nine regions experienced increased inequality, with an increase

ranging from about 0.6% to 6% (Fig. 3). Nevertheless, to ensure that the increase is not due to other demographic changes in this highly mobile population, such as changes in population size (Supplementary Table 3), we limited our analysis to regions ($n = 6$) where the increase in inequality score is above the 75th percentile, equivalent to an increase exceeding 2% ($\Delta Inq_v \geq .02$).

## Modelling multilevel networked outcomes

These data provide insights into varying social network configurations that reflect how rural populations maintain inter-household ties to sustain and improve their living conditions. For example, a shared physical environment is likely to reinforce the formation and maintenance of co-location social ties[20,53]. Considering the geographic barriers to accessing public facilities and the lack of public transport in our study context, these co-located social ties may be a major source of both insurance and influence on household resource allocation.

However, having cross-regional social ties can be seen as a crucial livelihood strategy under rural poverty – household members may locate in different regions in pursuit of diversification of income streams and with a means to collectively contribute to the socioeconomic welfare of their networked families[24,43,46]. Notably, many households rely heavily on remittances from members working in distant towns and cities[54]. These distant, cross-regional social ties have been suggested as a channel through which economic resources are transmitted across settings[25], thereby reinforcing a new form of labour economy through circular migration[22].

Here, we account for these regional-based social network structures and estimate their relative contribution to the distribution of economic resources. To do so, we link each household to an *isigodi* based on their geographic locations. This data construction method resulted in a bipartite, multilevel social network structure, with inter-household ties that may exist in the same *isigodi*, or across different *izigodi* (Fig. 4).

Figure 4 illustrates and describes the diverse set of small-scale, locally specified, multilevel social network structures linking households (circle, in blue) across *izigodi* (square, in red). We then build on recent advances of network methodology, namely the Autologistic Actor Attribute Model (ALAAM)[50,55], to simultaneously examine the association between outcomes observed at both levels, as well as how they may be associated with being embedded in varying

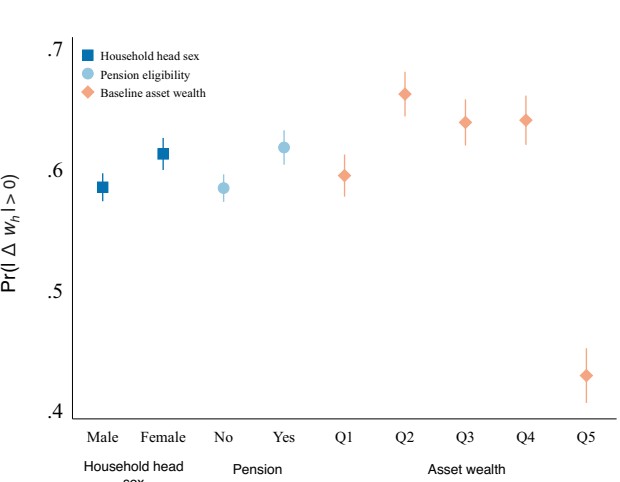

**a.**

**b.**

**Fig. 2 | Changes in household asset wealth from 2016 to 2018.** Panel **a** displays the overall proportion of households $h$ experienced an asset change ($|\Delta w_h| > 0$) versus a stable asset status ($\Delta w_h = 0$) across 23 regions (*izigodi*) from 2016 to 2018. In Panel **b**, we plot the average marginal estimates of the probability of a household

experiencing an asset change ($|\Delta w_h| > 0$) by their baseline attributes. These household-level attributes include household head sex (female/others), old-age non-contributory pension eligibility (whether the household has an aged 60-plus adult), and their baseline relative asset wealth status (in quintiles).

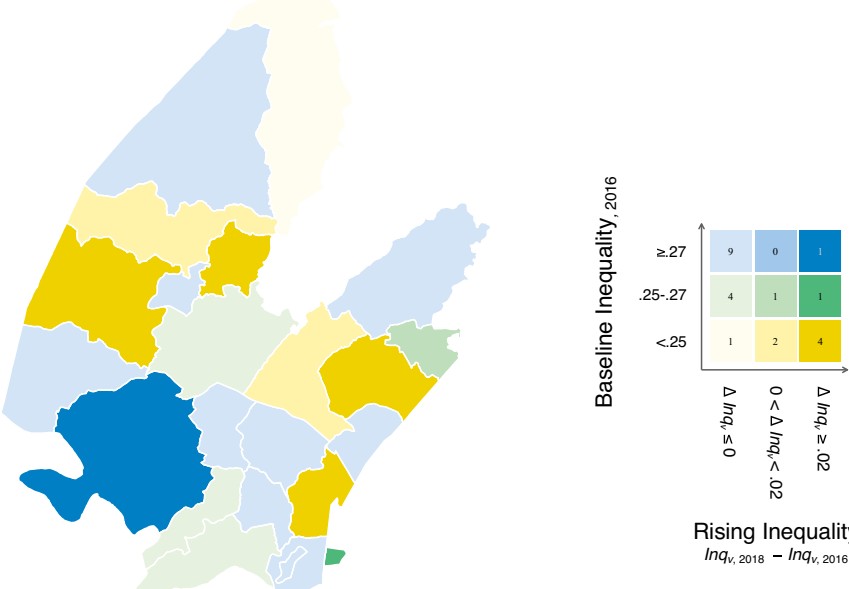

**Fig. 3 | Regional-level inequality outcome.** The figure displays the regional-level inequality outcome by *izigodi* ($n = 23$). We first compute a value indicating the percentage increase in regional-level $v$ inequality score over time ($\Delta Inq_v = Inq_{v,2018} - Inq_{v,2016}$). We then plot the distribution of this value against the regional-level inequality score at baseline ($Inq_{v,2016}$), categorising regions into those with a stable/decreased inequality score ($\Delta Inq_v \leq 0$), those with an increase ranging from about 0-2% ($0 < \Delta Inq_v < .02$), or those with an increase greater than 2% ($\Delta Inq_v \geq .02$). The value is then binarized as our regional-level outcome variable that indicates whether the area had experienced an increase in its inequality over time ($\Delta Inq_v > 0$). To ensure the increase is not driven by other demographic changes in this highly mobile population (e.g., population size), we restrict our analyses to regions ($n = 6$) with an increase in inequality score distributed at the upper 75th percentile ($\mu + (0.675)\sigma$), equivalent to more than a 2% increase ($\Delta Inq_v \geq .02$).

geographically defined social network structures (formally defined in 'Statistical analyses' in Methods).

## Network exposures and household economic outcome

Three principal findings emerged from our analyses that are associated with the allocation of household economic resources. First, our model revealed that socially and locally connected households are less likely to experience a change in their asset wealth status over time (Fig. 5a). The presence of a co-location social network tie reduces the odds for a household to experience an asset change by about 13%. In other words, by adding in one more social tie to a household in an *isigodi*, the probability for that household to experience an asset change decreases by a factor of about 0.14 more than we would expect at random, conditional on other household- and regional-level attributes and network parameters in the model. This finding highlights the potential of having local inter-household ties as a form of 'safety net' under pervasive poverty[14,15].

Second, we found that these co-location social ties are mutually reinforcing, indicating that a change in a household's economic condition is positively associated with a change in the economic condition of their network contacts (Fig. 5d). This co-location, local network interaction effect indicates that households are about 1.27 times significantly more likely to experience a change in their asset status if changes had also occurred in their network partners residing in the same region.

However, it should be emphasised that we are unable to demonstrate the casual relationships, as 'network interaction' here can be broadly defined as social processes via social network ties, including both social influence ('contagion') and social selection ('homophily'), that are related to our outcome of interest. In other words, the positive parameter suggests that the likelihood for a household to experience an asset change is being reinforced by asset changes of their network partners, either via the process of resource transmission (e.g., being

supplied resources, contagion), or the formation of new ties given prior living conditions (e.g., resource seeking, selection). In supplementary analyses, we found that these co-location network structural effects are sorted based upon one's initial asset endowments: Specifically, for the poorest households, having a network contact with the same asset status is correlated with an upward movement for both households (Supplementary Table 7).

Third, we found that circular labour migration remained an important livelihood strategy for rural households. Households are approximately 1.26 times significantly more likely to experience a change in their asset status by having a social tie outside of the DSA (Fig. 5c). However, this association follows an inverted U-shaped direction (Fig. 6). This implies a possible trade-off of having more working-age migrants on household resource allocation, as many of which may be obligated to care for all of the remaining household members, including children, orphans, and other dependants from families and relatives who are left in the rural home[18].

## Local network interactions and wealth inequality

Finally, it remains an open question as to how these inter-household network interactions could lead to an emergent phenomenon, influencing patterns of inequality observed at the regional level. Evidence from prior empirical models have suggested that micro-level social processes, often combined with homophily[56] (i.e., the tendency of individuals to associate with similar others) or high correlations between population traits[11] (e.g., low social mobility), could amplify various forms of intergroup inequalities. Studies on labour market outcomes[36,37] and information adoption[11,56,57] provide indications of such phenomena. Although lower-income populations often engage in economic activities with others in diverse geographic locations[22,24,25], prior network models have not adequately accounted for varying geographic boundaries embedded within the networked population. Furthermore, when estimating the

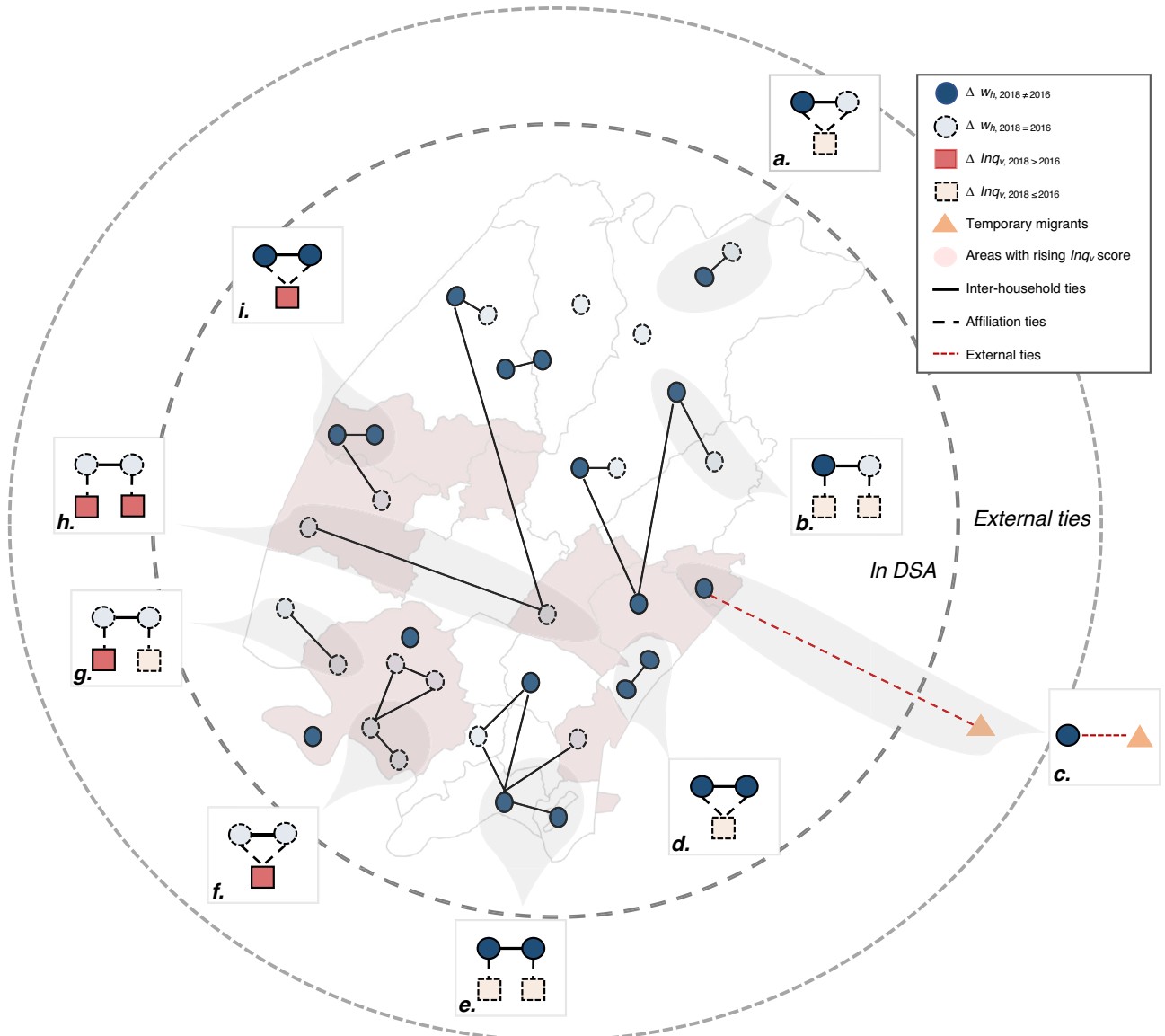

**Fig. 4 | Multilevel social network structures.** A graphical illustration of the key locally specified multilevel network structures in our network model, with inter-household ties existing within an *isigodi* and across *izigodi*. A darker colour indicates the household $h$ (circle, in blue) or the region $v$ (square, in red) has experienced an outcome change over time, whereas a lighter colour indicates others. We model three regional-based social connectivity effects for the household-level outcome: for example, the likelihood of a household experiencing an asset change ($|\Delta w_h|>0$) may be dependent on whether they have shared a tie to a household in the same *isigodi* (Fig. 4a), different *isigodi* (Fig. 4b), or outside of the DSA (external ties, Fig. 4c). External ties refer to households having non-residents at baseline (i.e., circular migrants). We model two regional-based network interaction effects, in which the likelihood of a household experiencing an asset change is dependent upon whether asset changes had also occurred among their network partners residing in the same *isigodi* (Fig. 4d) and/or different *izigodi* (Fig. 4e). For the regional-level outcome ($\Delta Inq_v \geq .02$), we examine whether rising inequality observed at the regional level is associated with their level of inter-household social connectedness (Fig. 4f), or by having many cross-region inter-household ties (Fig. 4g). Figure 4h indicates the likelihood of two interconnected regions (linked by inter-household ties) simultaneously experiencing a rise in their inequality scores over time. Finally, we introduce a cross-level interaction effect to examine whether the regional-level inequality outcome may be driven by network interaction processes within a shared geography – an indication of the theorised emergence of network-driven inequality (Fig. 4i).

micro-macro link, prior models have placed less attention to compare the relative importance of various network structures and mechanisms that may induce inequality[2,4].

Our model revealed that an increase in the regional-level inequality score is positively associated with local network interaction processes (Fig. 5i). Specifically, a rise in the inequality score is about 1.11 times more likely to be observed in regions where households are socially connected and economically interdependent. This parameter remains significant after adjusting for the effects of wealth-mobilised households in regions with rising inequality but without mutual connections – a parameter that is not statistically significant

('Cross-level interaction effect' in Fig. 7 and Supplementary Table 6). The robustness of the finding is further supported by alternative, more parsimonious model specifications (Supplementary Table 11). These analyses suggest that while social connections may span across multiple regions – notably given the high level of temporary residents of our study population – co-located social ties appear to be a major source of both insurance and influence on the patterns and dynamic of overall wealth variations.

Nevertheless, we recognise the possibility of reverse causality, where regional inequality patterns may also influence the formation and strength of local social interactions. To partly mitigate concerns

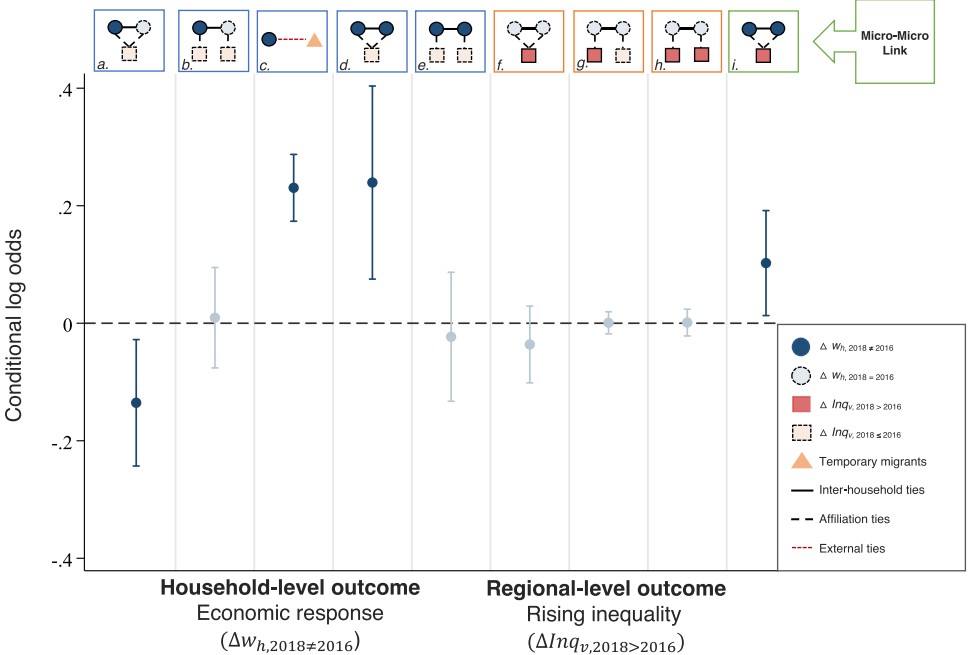

**Fig. 5 | Results from the multilevel ALAAM.** Parameters estimates are derived from the multilevel ALAAM. We have provided a detailed explanation for each network parameter (*a-i*) in Fig. 4. We present conditional log odds for the parameter estimates with 95% confidence intervals and highlight the statistically significant parameters in a darker colour. The model controlled for a range of baseline household- and regional-level attributes, as well as other network structural effects discussed in the Methods section. A positive parameter indicates that the observed, corresponding network configuration occurs more than we expect by chance, conditional on the rest of the estimated parameters in the model. Detailed parameter estimates, including all covariates are presented in Supplementary Table 6.

that changes in regional outcomes are due to other unobserved factors, such as changing population size, we compare the demographic composition across all regions and confirm a consistent population size (Supplementary Table 3). In this conclusion, our analyses may capture some degree of the network interaction effects associated with increased regional inequality against a backdrop of a stable population and after controlling for individual attributes, network dependencies, and structural effects.

## Discussion

Though efforts to reverse the effects of decades-long segregation have been underpinned through a system of progressive taxation, social assistance programs, and expansion of near-universal access to healthcare and education, South Africa remains one of the most unequal countries in the world, with severe poverty continuing to concentrate in rural black communities[58]. Building upon a long line of anthropological studies[13–15,18,43,45], our work provides a baseline understanding of the broad importance of one's primary social environments, including their social networks and geographic locations, on the allocation of household assets in shaping disparities in resource-ownership in a poverty-stricken setting. Findings here suggest that large-scale social intervention programs that account for the spillover potential of economic resources may be more efficient and effective than programs that ignore the importance of social connectedness in lower-income populations[31,38]. To this end, our work offers several avenues for future research.

First, our work provides a different data conceptualisation on how we could better understand the way in which the allocation of household resources, human behaviours, and health are socially patterned across interconnected rural communities on a larger scale. The sizable contribution of network effects observed in this study, particularly of localised network interaction, raises further questions as to whether it remains applicable to other socio-demographic outcomes, including family health and well-being. It also raises a question relating to how the rollout of large-scale, non-contributory social assistance

programs in South Africa, including the unconditional child support grant or old-age pension, may be better tailored to maximise the collective welfare of the entire networked families. Although our work is not without limitations (see 'limitations' in Methods), in many respects it enables follow-up studies to more precisely characterise the typology and magnitude of social connectedness that may influence the quality of rural domestic life.

Second, our analysis points to the value of considering co-existing network dynamics to better understand the causes and consequences of social inequalities. Concluding that socially connected families are better-off socio-economically than those who are less connected produces inaccurate theoretical and empirical remarks. As we have demonstrated in this work, a change in a household's economic conditions is associated with corresponding changes among their network peers in the same region, offering insights into the social contagion of economic resources. As such, under pervasive poverty, a socially connected setting may also exacerbate the degree of inequality by reinforcing network effects and resource dependencies. Given that levels of caring obligation and responsibility often increase synchronously with the levels of social connectivity in low-income settings[18], a question arising from our analysis is whether there remains a balance between social connections and the relative cost to maintain them. Our work therefore highlights the duality of social network effects[47], which may be better understood by considering the relative importance of various network mechanisms and structural arrangements.

Third, our work emphasises the utility of combining a stochastic, multilevel network modelling framework with population-level data to directly investigate how small-scale, locally-specified social processes and structures may combine to form global responses[28,48]. Continuing efforts to address social inequality have been largely based upon a top-down perspective, from increasing social spending to the readjustment of fiscal policies. These policies often imply a trade-off between increasing public spending for collective social welfare and its possible risk of debt distress. Across rural

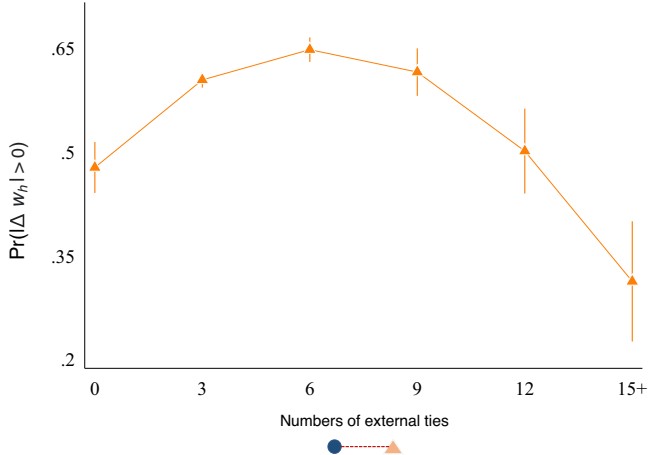

**Fig. 6 | The role of having temporary migrants on household asset wealth.** The figure displays the average marginal estimates on the probability for a household $h$ (circle, in blue) to experience an asset change ($|\Delta w_h|>0$) given the number of non-residents (triangle, in orange) at baseline (i.e., circular labour migrants, indicating those who do not live in the household for more than 6 months in a given year).

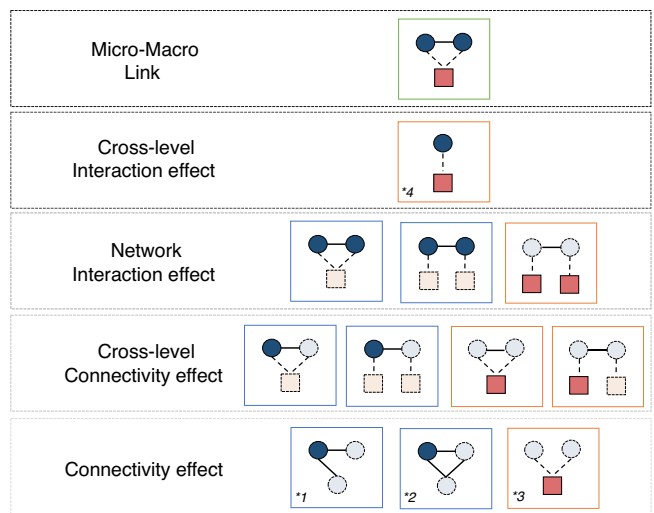

**Fig. 7 | Multilevel ALAAM specifications.** An illustration of the model building process in specifying multilevel ALAAM parameters. The figure displays all network parameters included in our final model. To begin, baseline structural effects, including a household-level connectivity effect (*1), closure (*2), and a regional-level connectivity effect (*3), are first added along with other nodal attributes. Following the illustrated sequence, we then proceeded the model specifications from the ground up. We note that the cross-level interaction effect (*4) measures the association between household- ($|\Delta w_h|>0$) and regional-level ($\Delta Inq_v \geq .02$) outcome change – a pre-condition for specifying the micro-macro link parameter (see Fig. 4i and Fig. 5i).

populations, however, a bottom-up perspective is urgently needed to characterise how resources are being re-allocated among interconnected families, the nature of these social relationships, and how variations in social assistance programs may help to alter these local social constraints. We hope our approach to traverse this micro-macro gap provides an avenue for further investigation on ways to understand inequalities from the ground up.

There may be other confounding factors that explain our findings, and we cannot prove the cause and effect between social network mechanisms, household resource allocation, and wealth inequality. In this study context, interpersonal relationships and social exchange are rooted in a range of complex historical local social dynamics that we cannot fully evaluate in this work. Nevertheless, observations from this study highlighted that network-centred strategies – such as those aimed at promoting resource sharing and support systems within social networks – may complement and potentiate the efficacy of social policy intervention designed to reduce disparities in economic opportunities across rural communities.

## Methods
### The study area
The Africa Health Research Institute Demographic Surveillance Area[39] (AHRI DSA), formerly the Africa Centre Demographic Information System[40] (ACDIS), was established in 2000 to monitor the health and socio-demographic development of an under-resourced, economically deprived, rural population with a high level of HIV prevalence to date[59]. Households are contacted three times annually to record information on births, deaths, and migration status of all previously and newly identified household members. The original DSA covered 438 km[2] in the uMkhanyakude district in the KwaZulu-Natal province, South Africa, and has since expanded to 845 km[2] with ~140,000 individuals in ~20,000 households by 2018. We organised the data as person-years and restricted our analysis to the original DSA.

As is typical for a rural South African setting, the DSA includes rural areas that were a designated Zulu 'homeland' during the apartheid era, and urban areas that formerly constituted a black-only township. While predominantly rural, the geographic areas are heterogeneous with respect to topography, density of settlement, and infrastructure development, in which residential units are scattered across areas, with no identifiable villages (hence, 'izigodi'). The primary source of income are waged labour and social grants.

## Constructing the multilevel social network
Given the unique social dynamics, poverty, and impact of population health factors like HIV, the conceptualisation of household in this population differs from other Demographic Surveillance and Community Surveys. AHRI DSA distinguishes between household, an indication of 'social group' in Zulu society (*umndeni*, household/family), and bounded structure, which is a physical place or residency (*umuzi*, homestead). Household membership is operationalised by asking "*who belong to this group*" rather than capturing "*where one stays*". Individuals can therefore have multiple, overlapping household memberships, but they can only reside in one homestead at any point-in-time. In Supplementary Note 1, we elaborate on the rationale behind this data conceptualisation in greater detail.

We primarily operationalised this indicator to construct the inter-household social network in 2016. Under this framework, two households are considered connected if they have nominated a common member from 2000 to 2016. However, this framework might not fully capture inter-household interactions involving more distant, weaker ties to extended families and other marital arrangements. For instance, the framework might not adequately account for situations like those of married women whose parents formally end their membership after receiving *lobola* (bridewealth) from the groom. Yet, the provision of care and support could still continue between the households[45,51,60]. As such, we also derived inter-household ties based on those who have changed their membership statuses over time. Detailed information on the demographic characteristics of respondents with multiple memberships can be found in Supplementary Note 1 and Supplementary Fig. 1. We further linked each household to an administrative unit (*isigodi*) based on their geographic location, resulting in a nested, multilevel social network structure. This approach captures potential inter-household ties within and across various *izigodi*.

## Household asset wealth
Monetary-based measures for household wealth in rural South Africa, such as income and expenditures, are often difficult to obtain and may

provide unreliable estimates of the total household wealth. This is in large part due to the highly seasonal and volatile nature of formal employment opportunities among rural populations, making it difficult to estimate the total monetary wealth accumulated by a household. Also, given a high unemployment rate of our study population (about 62% of the working-age population in 2018[39]), a substantial fraction of trade occurs as part of a barter or labour-sharing economy. In this case, asset-based measures are among the most commonly used and reliable estimates of household wealth, while also capturing a comparable underlying construct to a range of monetary-based measures[61].

We leveraged this asset-based index to construct our household-level economic outcome and regional-level inequality measure. The index was derived from the Household and Socio-Economic Survey (HSE) module available in 2016 and 2018, using Principal Component Analysis (PCA) on household assets items that consist of four broad subcategories – livestock, modern assets, water and sanitation access, and power supplies – for the entire observation period. The overall index was then organised into quintiles based on the first PCA component and was summarised as the relative asset scores of the two observed periods. The resulting quintiles range from 1 to 5, with a higher value indicating a wealthier household.

## Statistical analysis

We extend a new class of statistical network models to examine the correlation between various geographically defined network structural effects and economic outcomes both at the household- and regional levels. The method, called the Autologistic Actor Attribute Models (ALAAM), offers a principled analytical framework for estimating social influence processes that predict a binary outcome for cross-sectional social network data[50,55].

As an extension to the Exponential Random Graph Model (ERGM) – a class of social selection models ($p^*$models) that estimates the probability of a network tie between two nodes conditional on other network structural (e.g., 'preferential attachment'[26] and 'transitivity'[62]) and nodal attributes (actors' socio-demographics) – ALAAM models the probability of interdependent nodal outcomes while using network structure and other nodal attributes as covariates. It therefore shares similarities with logistic regression but does not assume statistical independence of the predicted outcome among observations (an actor's outcome can also be dependent on the outcome of other networked actors[50]). In this case, compared to other better-known statistical methods that primarily estimate a single network parameter (network autocorrelation models), ALAAM serves as an appealing approach to examine the correlation between network structures and social inequalities, as it affords direct modelling of a variety of dependency structures conceptualised in Fig. 4 that may give rise to a global, systemic economic change given random, locally-specified stochastic processes[55].

$$\Pr\left(Y_A = y_A, Y_B = y_B | A, B, X, Y_A^c, Y_B^c\right) = \frac{1}{\kappa(\theta_I)} \exp \sum_I \theta_I z_I(Y_A, Y_B, A, B, X, Y_A^c, Y_B^c) \quad (1)$$

Formally, the multilevel extension of ALAAM can be written as Eq. (1). Let $Y_A$ and $Y_B$ be the binary outcome for household-level ($A$) and regional-level ($B$) social networks, with cross-level interactions as ($X$) and other nodal attributes as $\{Y_A^c, Y_B^c\}$, respectively. Household-level attributes ($Y_A^c$) include household head sex (women/others), baseline asset status (in quintiles), mortality experiences (none/1 or more), old-age grant eligibility (whether the household has a 60-plus adult), external ties (number of non-residents, 0-15 +), and household size (0-20 + ), while ($Y_B^c$) is the baseline inequality score at the regional level (for further descriptive information, see Supplementary Note 2). We also adjusted for a continuous variable representing the

(mean) years in which households shared a tie through overlapping household members (0-16 years). Graph statistics are denoted as $z_I(Y_A, Y_B, A, B, X, Y_A^c, Y_B^c)$ that count the number of different network configurations illustrated in Fig. 4. The estimated parameters, $\theta_I$, determine the relative contribution of various network configurations on the outcomes at both household ($A$) and regional ($B$) levels, $\{Y_A, Y_B\}$. Hence, a positive estimated parameter of a network configuration suggests the corresponding configuration occurs more than we expect by chance, conditional on the rest of the estimated parameters in the model. $\kappa$ is a normalising constant, ensuring a proper probability distribution. We obtain all estimated parameters by employing the Markov Chain Monte Carlo (MCMC) maximum likelihood methods[63] implemented in the MPNet software[64].

## Analytical strategy

The model specification in ALAAM follows a general hierarchy of complexity[49], from lower-level nodal attribute effects (socio-demographics) to higher-level multilevel network structural effects (Fig. 4). Our analysis therefore proceeded from the ground up to ensure our final models were not under- or overparametrized. We first assessed the descriptive patterns of household- and regional-level outcomes across *izigodi*. We ran a series of logistic regressions predicting the likelihood for a household to experience an asset change by their baseline attributes in 2016 (Fig. 2), as well as the likelihood of an area to experience a rise in inequality score based on their baseline inequality score (Fig. 3). These results are discussed in greater detail in Supplementary Note 3.

We next extended the current ALAAM specifications to account for the multilevel, social-geographical dependency structures. In specifying ALAAM parameters, baseline structural factors, including density (akin to constant) and lower-level structural indicators, are generally included as pre-conditions before the specification of higher-order structural effects. Here, we illustrate such model specifications in Fig. 7. For example, a higher-order structural effect like the cross-level connectivity effect should be included after the specification of density and lower-level household- and regional-level connectivity effects. Network interaction effects are also specified along with the specification of single-level and cross-level connectivity effects. We therefore followed a step-wise approach by first adding in nodal attributes, connectivity-based network effects, then to higher-order multilevel social-geographical network effects. Figure 7 presents all network parameters included in our final model.

## Model fits

There is a large number of possible network configurations in the multilevel ALAAM specification. For configurations that involved network ties ($A, B, X$), we mainly focused on network configurations related to our theoretical interests illustrated in Fig. 4 to ensure our final models were not overparametrized. However, given various possible combinations of $\{Y, A, B, X, Y^c\}$ in $z_i$, there may be other graph statistics that represent features of the observed, population-level data. We thus assessed the goodness of fit (GOF) statistics by simulating all implemented graph statistics in the MPNet software for the converged model[49,65]. This modelling procedure is equivalent to that in ERGM, which compares observed statistics with simulated samples from the converged model using $t$-ratio, with a value smaller than 1.96 indicating model convergence and an adequate representation to the observed data[49,55,65].

To improve model GOF so that all statistics implemented under MPNet are adequately captured, we subsequently included a selection-based network interaction effect for the poorest households. This result suggests that network effects may be sorted based on a household's initial economic position, specifically among the poorest households (Supplementary Table 7).

## Limitations

Social relations are inherently and invariably complex, and our findings need to be interpreted with caution. First, our work may not be generalised to other settings given the unique social and historical local dynamics of our study population. However, given that a majority of these inter-household connections are kin-based, there is likely benefit to extend the network construction method to other settings, especially in settings with strong familial ties (e.g., in parts of South America, Asia, or other sub-Saharan settings). Although further research in other settings is important, we instead point to the importance of considering the utility of social protection grants under varying network structures across lower-income contexts (e.g., co-located network ties), as prior evidence is mostly derived from treating a household as an independent unit of analysis[66,67]. Second, there may be other unobserved factors that are driving our findings, and we are unable to claim causality between social network effects and our outcomes. Nevertheless, as we have demonstrated in this study, some endogeneities are arguably the main interest of network research, including the inherently confounded network mechanisms between selection, influence, location, and network dependencies[68]. Fortunately, ongoing methodological efforts have created opportunities to address these challenging issues of causality in networks for longitudinal data[69,70] and we encourage more works applying such an approach to contextualise the interdependency of rural domestic life. Finally, our findings are only applicable to the observed population. We are thus unable to measure possible network effects among the group with missing records on the outcome variable ($n = 1160$). In addition to other robustness tests that we have further discussed in Supplementary Note 3, we also report the demographic characteristics of the population to whom our results most directly generalise. These descriptive patterns suggested that those with missing information appears to be less socially connected and without the stable financial support from the old-age grant (Supplementary Table 12).

## Reporting summary

Further information on research design is available in the Nature Portfolio Reporting Summary linked to this article.

## Data availability

Data are available in a public, open access repository. The data underlying the results presented in the study are available from the AHRI Data Repository (https://data.ahri.org). The raw data is published on the data repository as licenced datasets. To access the data, users must self-register on the data repository and click through a data use agreement and submit their request.

## Code availability

The R, Stata, and MPNet code used to generate the results is available from the corresponding author upon request.

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

## Acknowledgements

We thank all the respondents who participated in the census. We are also indebted to the field teams and the AHRI HDSS data management teams for collecting, managing, and providing the data. The Africa Health Research Institute (AHRI) is funded by the South African Department of Science and Innovation through the South African Population Research Infrastructure Network (SAPRIN), which is hosted by the South African Medical Research Council. AHRI receives core funding from the Wellcome Trust (Grant number: 201433/Z/16/Z) for the aspects of its health and demographic surveillance. STY is supported by the ANU-Taiwan PhD Scholarship from the Ministry of Education, Taiwan. GH is supported by a fellowship from the Wellcome Trust and Royal Society [Grant number 210479/Z/18/Z]. This research was funded in whole, or in part, by the Wellcome Trust [Grant numbers 201433/Z/16/Z and 210479/Z/18/Z]. For the purpose of open access, the author has applied a CC BY public copyright licence to any Author Accepted Manuscript version arising from this submission.

## Author contributions

S.T.Y. designed the study, conducted the analysis, and wrote the first draft of the manuscript with support from B.H., P.W., C.W.K., D.G., and G.H. D.G. and G.H. contributed to all stages of data collection and management. All authors reviewed and approved the final version of the manuscript.

## Competing interests

The authors declare no competing interest.
