## [Peer Review File · Nature Communications]

Local Network Interaction as a Mechanism for Wealth InequalityReviewers' Comments:

Reviewer #1:

Remarks to the Author:

This paper, entitled "Local Network Contagion as a Mechanism for Wealth Inequality," scrutinises the effects of inter-household connections on asset conditions and changes in wealth inequality at an aggregated level. Based on their network and their results, the authors find that the dyadic structure of the household network significantly influences income variation. I concur with the authors regarding the importance of this research topic, particularly how social capital and its structure impact wealth distribution and dynamics, especially in the context of developing countries.

I am also impressed by the dataset the authors used. This unique household membership database allows the researchers to examine theories in the context of a country where such knowledge is even more needed. Moreover, I perceive that a significant majority of academic research is written about or from the context of Western developed societies. So, I appreciate the authors' effort to conduct their research in a region of a more challenging-to-research country. I think, however, that the authors could demonstrate their hypothesis and results in a more effective and clearer way. I am not sure if I correctly understand the argumentation and figures in the paper because of the lack of detailed clarification, especially mainly from a statistical and network science perspective. Thus, I regret not being able to ascertain whether the analysis properly supports the paper's findings and conclusions.

The introduction to the paper adequately summarises the requisite literature for interdisciplinary research in this field, but when it comes to justifying its own motivation, the paper runs into inconsistent statements. The introduction says that "Survey-based studies, which are the most commonly employed approach for such investigations, are often prone to inaccurate self-report responses and can be costly at scale." I don't really understand this type of motivational reasoning as the authors also rely on data filled in by the household members themselves. Who they think is a household member or head is not administratively generated data but self-reported. When I read this section in the introduction, I expected they had found a source that could identify household members without reporting bias.

In addition, the authors say that their research fills a gap because in previous research "[I]dentification of a causal network effect, either via natural experiment or a randomised controlled trial, provides valuable insights into the causes and consequences of social networks by ruling out endogeneity. These approaches, however, are limited in their ability to bridge the micro-macro gap, as they mainly reveal the relationship between macro-level inequality outcomes and global network properties or individual-level economic outcomes and local social interactions." Although, in fact, the authors do nothing more than try to compare network motives with household and regional outcomes. Only their statistical techniques do not even handle endogeneity or identify causality since the interconnection of households is clearly not independent of their prior financial situation and vice versa. Moreover, reading the results and the methodology section after the introduction, I got the impression that the two sections were written by two different people without any agreement on what exactly their methodological tools were capable of and why they were doing this type of network analysis.

The construction of the network is also not clearly elucidated and not well explained. I had to go through it four times before I understood why I didn't understand it before. The article lacks a part that describes in a clear way the generative process behind the construction of the network. Even after repeated reading, it is not clear whether, in addition to the household-household pair, the network's descriptive statistics include regional ties or whether only the household-household unipartite graph is analysed in that section.

If we only look at the network projected onto households, are the points in the same geographical region connected automatically because of the bipartite structure? If so, does this not cause a lot of

artificial binding between households? If not, what is the added value of the geographical bipartite projection here? If households are connected together just because they are in a region, then we can indeed talk about a multilevel network. However, then all households in the same geographical unit will be artificially connected, generating many false connections. This is partly explained by the authors by the fact that all households in a region belong to the same community. However, there are many methods for detecting real communities in network science that can find local tightly knit groups, just like the Louvain method or Girvan-Newman algorithm. Considering everyone in the same geographical unit part of the same community is a way too assumption.

However, if the authors do not generate a link between two households just because they are in the same region, then we cannot really talk about a multilevel network. Instead, they put two bipartite structures next to each other, looking at the projection of one of the unipartite ones. This problem is not clarified anywhere in the paper.

There are also many tools for analysing spatiality, from spatial econometrics models to the use of different weight matrices in network construction. However, I feel that the authors are currently overcomplicating the analysis and incorrectly treating geographical regions as social communities.

Missing from the paper is a graph visualising of the network. From the very limited descriptive data we see of the network, we do not know whether it is close to a random, possibly scale-independent, or small-world network, or whether due to its uniqueness, we are seeing a completely different type of structure. Preliminarily, I would think that the visualisation of the network would show a maximum-spanning-tree-like structure that should be treated differently than most social networks. Still, we do not know this due to the lack of visualisation and descriptive statistics.

I also had a major conceptual problem with why they are looking at the absolute value of the asset change (let's ignore the fact that they are looking at two principal components calculated at different times which are hardly comparable or interpretable; moreover, we don't really know the details of how many components and how strongly they are correlated with the main variable). If they look at the absolute value of change, we don't really know if certain network motives are good for the households, since the "good" status will be reflected if they have not increased or decreased but maintained their position. The authors probably did this because they needed a binary dependent variable for the ALAAM method. However, they could have easily solved this methodological problem by testing the network with different alternative specifications:

- 0 no change, 1 positive change;
- 0 no change, 1 negative change;
- 0 decrease, 1 increase.

Finally, I did not find any sense in the multinomial 3-way interaction. This may be my mistake, but multinomial regressions are difficult to interpret correctly without adding triple interactions with squared terms making all the assumptions inconceivable. Based on the dependent variable, this should be an ordered logit regression.

Reviewer #2:

Remarks to the Author:

I thought this was an interesting paper, applying a complexity approach to the enduring question of wealth inequality in South Africa. The methodological approach seems sound and well-justified to me. I think the methodology employed would be very useful to other researchers in the field examining related question, particularly given that there are a number of HDSS sites around South Africa, opening up the possibility for comparative work across and between the sites.

My main comment is about the presentation of the material, which leaves the reader with some

questions which are addressed only later on in the paper about the data, the type of inequality, and some other issues. I would suggest a brief discussion of these earlier on in the piece, possible in the introduction. I would suggest more detail on what is meant by "ownership of resources", the location of the study site, and what the authors understand by the term "emergent". This will help an unfamiliar reader better locate the study.

Similarly, I think the study could be better located in the South African literature in the introduction. A nod is made to the anthropological lineage into which the study fits, but a brief elucidation of this would be useful.

Some aspects of the discussion are opaque. For example, on page 15 the authors state "Nevertheless, observations from this study give us the confidence to hypothesise that network-based efforts to address the negative social consequences under rural poverty will more realistically and effectively achieve the desired outcomes derived from social policy interventions." It is not clear what this means. What are the "network-based effects to address the negative social consequences"? What are the effects, how could they be used to address, and what are the negative social consequences? What sort of social policy interventions are conceived of?

Reviewer #3:

Remarks to the Author:

This paper, "Local Network Contagion as a Mechanism for Wealth Inequality," examines the links between micro-level social networks and macro-level wealth inequality using South Africa's large survey data. I appreciate the authors' technical approach to examining the links between micro-level social networks and macro-level wealth inequality with the large-scale empirical dataset.

However, I believe that the authors could improve the paper by addressing the following concerns:

First, I suggest that the authors showcase the reproducibility of their analysis in greater detail. The paper's central argument relies heavily on the results presented in Figure 5, particularly Figure 5i. Therefore, the authors need to demonstrate how robust their findings are, even when using various settings in the statistical model. While the authors touch upon their "robustness check" in Supplementary Note 3, they could conduct additional analyses to show how many errors in the data could be tolerated to arrive at the same conclusion drawn from the analysis. Furthermore, it is essential to test the impact of missing data and the possibility of missing controlled parameters in the statistical results. The authors could show robustness against the missing data or parameters by randomly eliminating a certain percentage of households or parameters from their dataset.

Second, I suggest that the authors explain the statistical model and the interpretation of results more clearly. The multilevel network analysis shows whether a "network configuration occurs more than we expect by chance, conditional on the rest of the estimated parameters in the model," and all the results from the model indicate associations and not causalities. Although the authors have clarified this point in the manuscript, some interpretations suggest causal relationships. For instance, in the abstract, the authors use "support," "promote," and "contribute to an increase" to describe their findings, but such terms imply causal directionality, which is misleading for their association analysis. If the authors can justify their causal claims from their analysis, they should elaborate more on them. Otherwise, the authors should avoid using terms that imply causal directionality over the paper and instead focus on the associations revealed by their analysis.

Related to this, I suggest that the authors clarify the statistical model even in the main text. In my first reading, I did not clearly understand that each network composition shown in Figure 5 is nested in the analysis, as shown in Figure 7. My confusion came from what the light-colored (circle and square) nodes indicate in Figure 5. Now I understand the light-colored nodes mean "unspecified"; i.e.,

for example, Fig. 5i's composition is nested in Fig 5d's one, and Fig. 5d's one is nested in Fig 5a's one. However, in Figure 4, the authors seem to use the light-colored nodes for "no change" in opposition to the dark-colored nodes; in that case, Fig 5i's composition should complement Figure 5d's. The authors should clarify this point to avoid possible confusion.

Third, I recommend that the authors provide more nuanced explanations of the association link to clarify the study's importance. Community-level wealth inequality is calculated based on the accumulation of individual wealth. Thus, when the households in a community have no changes in wealth, the community's wealth inequality has no way to increase or decrease for calculation. The main result shown in Fig. 5i ostensibly confirms the apparent micro-macro association. I think their analysis should provide more nuanced explanations, but they are unclear. For example, does Figure 5i's composition still hold the significant odd even controlling the composition where wealth-mobilized households individually belong to inequality-increased communities without connections between them? If so, the network effect would be highlighted. Furthermore, from the perspective of addressing the societal problem, it could be more important to show how network mechanisms help people reduce wealth inequality rather than increase it. Thus, It would be beneficial to investigate whether the result would be just the opposite of Figure 5 if the community-level binary variable is set in the opposite way.

Finally, I would suggest that the authors describe the people who connect the households (or "social groups") in more detail, e.g., by giving examples and showing the percentages in a table. The interpretation of a network analysis depends on the substance of the network links (or edges), and the network effect could differ according to the link types. Thus, such additional analyses would help readers imagine how the bridging people work on economic mobilization in the specific field and context. They would also help consider how this study's findings could be applied to other populations.

Minor comments:

- In Figure 4, the authors colored the areas where the rising inequality scores are more than 0 but mentioned the areas where they are more than 0.02 in the caption.
- Figure 2a lacks the labeling of the light-colored part.
- The authors use "isigodi" and "izigodi" for the same meaning, including the figures. Please unify the terms.

Reviewer #1 (Remarks to the Author):

[1] This paper, entitled “Local Network Contagion as a Mechanism for Wealth Inequality,” scrutinises the effects of inter-household connections on asset conditions and changes in wealth inequality at an aggregated level. Based on their network and their results, the authors find that the dyadic structure of the household network significantly influences income variation. I concur with the authors regarding the importance of this research topic, particularly how social capital and its structure impact wealth distribution and dynamics, especially in the context of developing countries.

[Reply 1] To begin, we would like to thank the Reviewer for all the engaged, critical, and constructive comments on our manuscript. They have certainly helped to strengthen both the conceptual and empirical aspects of this work. In this revised manuscript, we have accordingly focused on further refining our clarity in writing, explaining our network construction method, and detailing the population-scaled inter-household social network. Below, we address each comment individually, and we thank the reviewer for taking the time to review this work.

[2] I am also impressed by the dataset the authors used. This unique household membership database allows the researchers to examine theories in the context of a country where such knowledge is even more needed. Moreover, I perceive that a significant majority of academic research is written about or from the context of Western developed societies. So, I appreciate the authors’ effort to conduct their research in a region of a more challenging-to-research country. I think, however, that the authors could demonstrate their hypothesis and results in a more effective and clearer way. I am not sure if I correctly understand the argumentation and figures in the paper because of the lack of detailed clarification, especially mainly from a statistical and network science perspective. Thus, I regret not being able to ascertain whether the analysis properly supports the paper’s findings and conclusions.

[Reply 2] The Reviewer has rightly pointed out that most existing work on the relationships between social networks and income variations is based in higher-income settings. We believe this limitation is largely due to the lack of network data that enables research in rural, lower-income, and ‘challenging-to-research’ contexts. We also note that existing methodologies for constructing population-scale network data, such as digital

trace approaches, may not fully capture the scope of social interactions in these settings. As such, our work focuses on devising potential alternatives that may facilitate research in investigating network effects in such settings. Nevertheless, given how our network data is constructed, especially in the context of our study population, we concur with the Reviewer on the need for a more detailed and elaborate description of the data construction method and motivations. In the following responses, we therefore describe how we revised many parts of the manuscript to address each of the concerns raised.

[3] The introduction to the paper adequately summarises the requisite literature for interdisciplinary research in this field, but when it comes to justifying its own motivation, the paper runs into inconsistent statements. The introduction says that “Survey-based studies, which are the most commonly employed approach for such investigations, are often prone to inaccurate self-report responses and can be costly at scale.” I don’t really understand this type of motivational reasoning as the authors also rely on data filled in by the household members themselves. Who they think is a household member or head is not administratively generated data but self-reported. When I read this section in the introduction, I expected they had found a source that could identify household members without reporting bias.

[Reply 3] The reasoning behind the highlighted sentence aimed to illustrate how one might approach the challenges of scaling social network data in poorer, remote settings, particularly through alternative solutions. In our case, we have leveraged a context-specific measure of overlapping household memberships derived from census data to map inter-household relationships. Nevertheless, we agree that this justification needs to be better motivated and more clearly elaborated. We have revised the entire paragraph as follows:

“First, limited studies have been conducted in rural and low-income populations^{2,4}. Previous anthropological research has highlighted the importance of social networks as critical support systems in poorer environments¹³⁻¹⁵. Yet, empirical investigation exploring the influence of social networks on inequalities in these settings pose significant scalability challenges. Survey-based approaches, while common, often encounter limitations in remote populations, hampered by high costs and inadequate infrastructure for extensive research activities. As a result, existing research tend to focus on specific population sub-groups¹⁶, lacking data on network interactions

observed at the population level. In response, computational and online experimental techniques may offer scalable alternatives^{7,9,11}. However, it remains in question whether these digital proxies can capture more tangible, costly, and culturally-defined offline behaviours related to resource-sharing and exchange activities among families in poverty-stricken settings^{17,18}. The availability of population-level in-person data therefore offers an important opportunity to gain a deeper understanding of the under-researched correlation between social networks and economic inequality in rural, poorer contexts.”

[4] In addition, the authors say that their research fills a gap because in previous research “[I]dentification of a causal network effect, either via natural experiment or a randomised controlled trial, provides valuable insights into the causes and consequences of social networks by ruling out endogeneity. These approaches, however, are limited in their ability to bridge the micro-macro gap, as they mainly reveal the relationship between macro-level inequality outcomes and global network properties or individual-level economic outcomes and local social interactions.” Although, in fact, the authors do nothing more than try to compare network motives with household and regional outcomes. Only their statistical techniques do not even handle endogeneity or identify causality since the interconnection of households is clearly not independent of their prior financial situation and vice versa. Moreover, reading the results and the methodology section after the introduction, I got the impression that the two sections were written by two different people without any agreement on what exactly their methodological tools were capable of and why they were doing this type of network analysis.

[Reply 4] Indeed, our approach does not fully address endogeneity concerns nor does it establish causality. We aim to highlight that many existing studies tend to indirectly infer population-level phenomena from micro-level observations or vice versa. These studies often overlook the *complexities* of accurately translating micro-level behaviours into macro-level outcomes—particularly, how various predicted social patterns and the randomness in human interactions may jointly operate. Such processes may complicate the micro-to-macro transition. Existing network research on social inequality, however, tends to focus on either micro- or macro-level analysis.

One advantage of ALAAM, particularly when including multilevel features, is its ability to jointly model how household- and regional-level outcomes may be

interdependent, as well as how various structural and compositional effects may correlate with these outcome changes—given random processes. In this case, we thank the Reviewer for pointing out such redundancy in our writing, and we have accordingly revised the paragraph, which now reads as follows:

“Third, several longstanding questions in social and natural sciences relate to the puzzle of whether small-scale social processes can lead to emergent phenomena at the macro level²⁶⁻³⁰. This micro-to-macro question, pertinent in understanding phenomena such as the recent COVID-19 pandemic³¹ and climate change responses³², seeks to clarify how local social interactions might traverse into population-level dynamics. However, understanding these micro-macro linkages remains a formidable challenge, often due to the methodological complexities involved in harmonising and modelling the expected social patterns and the inherent randomness in human social interactions³³. Previous research has explored the association between macro-level inequality and global network properties^{7,34}, as well as individual-level economic outcomes and local social interactions³⁵⁻³⁷. Yet, the link between macro- and micro-level economic outcomes, alongside the specific types of social interactions that may strengthen returns capable of influencing economic disparities, remains poorly understood. Unpacking the varying network dynamics at play is therefore likely to contribute to a more refined understanding of how to enhance the effectiveness and efficiency of social policy interventions^{31,38}.”

[5] The construction of the network is also not clearly elucidated and not well explained. I had to go through it four times before I understood why I didn't understand it before. The article lacks a part that describes in a clear way the generative process behind the construction of the network. Even after repeated reading, it is not clear whether, in addition to the household-household pair, the network's descriptive statistics include regional ties or whether only the household-household unipartite graph is analysed in that section.

[Reply 5] We believe the previous structure of our manuscript may have contributed to this confusion, and that our network descriptive statistics do not include regional ties. To clarify our data construction method, we have reorganised the results section as follows: First, we describe how these inter-household ties are formed and the motivations behind these relationships based on existing observational works—that is, under what

circumstances two households would nominate the same member(s). Second, we present basic descriptive statistics to help contextualise our network data. This is accompanied by a newly added stylised figure to illustrate the data construction method (Fig 1a), along with a network figure that visualises the sub-structures of the inter-household social network (Fig 1b and Fig 1c). The text now reads as follows:

“Our study population comprises 11,834 households (92,688 people) observed in 2016, scattered across 23 *izigodi*³⁹. To construct the social network for this population, we integrate a prospective, detailed household residency survey spanning from 2000 to 2016 to identify inter-household ties based on overlapping household members. Here, we consider two households as connected if they have shared a common member in the study timeframe (Fig. 1a). AHRI DSA distinguishes one’s household memberships (household) with their actual residency status (homestead) to capture the ‘stretched households’ arrangements observed across many rural South African settings, whereby families work together socio-economically while living apart⁴³. In this population, a concurrent membership reflects inter-household ties that are predominantly driven by a combination of shared responsibilities (care receiving and giving), authority (headship and polygamy), and identity (kinship) between households, as well as by historical social relations that may give rise to a sense of belongingness between families^{41,43,45}. In these instances, household members are thus usually, but not always, related.

[Figure 1 about here]

Descriptively, this sparsely connected undirected network consists of 10,162 inter-household ties and follows a long-tailed degree distribution with an average degree of 1.72 (SD = 1.8) per household (Fig. 1b). This network has a global clustering coefficient of approximately 0.21 and an average path length of about 11.84 (SD = 2.98). Over 80% of households, approximately 8,453 in total, are connected through a shared member. The largest connected component in this network consists of about 6,462 households (Fig. 1c). Further information on the setting, data conceptualisation, and network construction are discussed in greater detail in the Methods section and Supplementary Note 1.”

[6] If we only look at the network projected onto households, are the points in the same geographical region connected automatically because of the bipartite structure? If so, does this not cause a lot of artificial binding between households? If not, what is the added value

of the geographical bipartite projection here? If households are connected together just because they are in a region, then we can indeed talk about a multilevel network. However, then all households in the same geographical unit will be artificially connected, generating many false connections. This is partly explained by the authors by the fact that all households in a region belong to the same community. However, there are many methods for detecting real communities in network science that can find local tightly knit groups, just like the Louvain method or Girvan-Newman algorithm. Considering everyone in the same geographical unit part of the same community is a way too assumption.

[Reply 6] Households in the same region (*isigodi*) are not considered connected automatically. Only if both households have nominated a common member do we consider them to be connected. In this case, inter-household ties may exist in the same region (*isigodi*), or across multiple regions (*izigodi*). Again, we believe such confusion is largely due to the way in which we have not yet effectively communicated our data construction method, study motivations, and from our use of the term ‘community’ to indicate a region.

We are primarily motivated to investigate how social connections across various regions may influence household and regional inequality, given high rates of internal and international labour migration in the Global South, especially in rural South Africa. In demographic research, scholars have highlighted the importance of these distant economic ties for family livelihood in poorer environments (e.g., *The New Economics of Labour Migration*). Economic research has also shown how migration networks could influence the level of income variation in the sending country. In fact, anthropological works have long highlighted the key role of distant family ties in household livelihood and economic well-being. Moreover, sociological studies have demonstrated how global-level network structures in a region is associated with its income inequality in higher-income contexts. As such, with empirical evidence being limited in poorer settings, our aim is to extend these discussions by examining how varied social network ties, which cross geographic boundaries, might relate to economic outcomes at both household and regional levels, especially within the highly mobile, rural, poorer South African contexts.

In line with the previous comment, we have reorganised our results section to first introduce the inter-household social network data, then present basic descriptive statistics for this network and our outcome and control variables. Next, we further detailed the motivations for examining social ties across different regions and their potential impact on wealth variations. To clarify terminology, we have substituted ‘community’ with ‘region’.

To begin with, we have revised the introduction to more effectively justify the theoretical motives:

“Second, most existing research has overlooked the intricate and multi-layered network structures observed in real-world social networks, including one’s affiliation in groups¹⁹, residency in places^{19,20}, and their connections to ecological spaces²¹. The accelerated urbanisation and globalisation of modern societies have altered the ways in which people are connected²², particularly in the Global South²³. This pattern is evident among rural populations with limited job prospects, whose livelihoods rely primarily on economic support from distant sources²⁴. These physical and geographic boundaries may have a considerable influence on a range of economic outcomes for the interconnected individuals²², as well as the prosperity of these communities at large^{7,25}. Nevertheless, little is known about the degree to which these network interactions—spanning across diverse geographic boundaries—are associated with the economic development of rural communities. As such, an alternative, multilevel social network framework is required to better estimate the effect of these geographically defined network structures on social inequalities.”

[7] However, if the authors do not generate a link between two households just because they are in the same region, then we cannot really talk about a multilevel network. Instead, they put two bipartite structures next to each other, looking at the projection of one of the unipartite ones. This problem is not clarified anywhere in the paper. There are also many tools for analysing spatiality, from spatial econometrics models to the use of different weight matrices in network construction. However, I feel that the authors are currently overcomplicating the analysis and incorrectly treating geographical regions as social communities.

[Reply 7] Following the previous comment, we have revised and reorganised parts of the results section to justify our aims to examine the associations between various geographically defined network structures and their effects on wealth dynamics:

“These data provide insights into varying social network configurations that reflect how rural populations maintain inter-household ties to sustain and improve their living conditions. For example, a shared physical environment is likely to reinforce the

formation and maintenance of co-location social ties^{20,53}. Considering the geographic barriers to access public facilities and the lack of public transport in our study context, these co-located social ties may be a major source of both insurance and influence on household resource allocation.

However, having cross-regional social ties can be seen as a crucial livelihood strategy under rural poverty—household members may locate in different regions in pursuit of diversification of income streams and with a means to collectively contribute to the socio-economic welfare of their networked families^{24,43,46}. Notably, many households rely heavily on remittances from members working in distant towns and cities⁵⁴. These distant, cross-regional social ties have been suggested as a channel through which economic resources are transmitted across settings²⁵, thereby reinforcing a new form of labour economy through circular migration²².

Here, we account for these regional-based social network structures and estimate their relative contribution to the distribution of economic resources. To do so, we link each household to an *isigodi* based on their geographic locations. This data construction method resulted in a bipartite, multilevel social network structure, with inter-household ties that may exist in the same *isigodi*, or across different *izigodi* (Fig. 4).”

[9] Missing from the paper is a graph visualising of the network. From the very limited descriptive data we see of the network, we do not know whether it is close to a random, possibly scale-independent, or small-world network, or whether due to its uniqueness, we are seeing a completely different type of structure. Preliminarily, I would think that the visualisation of the network would show a maximum-spanning-tree-like structure that should be treated differently than most social networks. Still, we do not know this due to the lack of visualisation and descriptive statistics.

[Reply 9] We thank the Reviewer for highlighting this and have incorporated a new figure visualising the largest connected component and the sub-structures of this component (Fig. 1). We have also included a table in the supplementary materials to showcase basic descriptive statistics of the inter-household social network, without regional ties (Supplementary Table 1):

Metric	Value	Notes
$\langle k \rangle$	1.7174	Average degree (mean number of connections per node), SD = 1.7966.
C_g	0.2077	Global clustering coefficient, indicating the overall likelihood that two neighbours of a node are connected.
$\langle C \rangle$	0.3032	Average local clustering coefficient, indicating the average probability that two neighbours of a node are connected. SD = 0.3769
$\langle l \rangle$	11.8418	Average path length, indicating the average number of steps along the shortest paths for all possible pairs of network node, SD = 2.9752
N	11,834	Total number of nodes in the network.
E	10,162	Total number of edges representing inter-household ties.
ω	4,068	Number of connected components in the network.
N_{max}	6,462	Number of nodes in the largest connected component of the network.

[10] I also had a major conceptual problem with why they are looking at the absolute value of the asset change (let's ignore the fact that they are looking at two principal components calculated at different times which are hardly comparable or interpretable; moreover, we don't really know the details of how many components and how strongly they are correlated with the main variable). If they look at the absolute value of change, we don't really know if certain network motives are good for the households, since the "good" status will be reflected if they have not increased or decreased but maintained their position. The authors probably did this because they needed a binary dependent variable for the ALAAM method. However, they could have easily solved this methodological problem by testing the network with different alternative specifications:

0 no change, 1 positive change;

0 no change, 1 negative change;

0 decrease, 1 increase.

[Reply 10] The Reviewer has rightly pointed out that the choice to binarize our outcome is primarily due to the nature of ALAAM. We also agree with the Reviewer that there may be other important aspects and approaches to contextualising household wealth indicators, and that we were largely leveraging the data and measure available to us when this study was conducted. From the study site, PCA scores were calculated for the entire observation period, and we constructed the relative quintiles and changes using data from within our study timeframes. However, in addition to our previous supplementary analyses—in which we categorised the outcomes as either positive (upward) or negative (downward) changes—we have now run an additional analysis to indicate 'no change', with any change as the reference group. Results can be found in Supplementary Table 7, 8, and 9, and are further elaborated in Supplementary Note 3 as follows:

“...Results from these models suggest that households are about 1.4 times ($e^{0.3390}$) significantly more likely to move downward if the same change occurred in their co-located network partners (Supplementary Table 7). Having social ties to those who are outside of the DSA is associated with an upward movement (Supplementary Table 8); and that socially and economically contagious upward movements are about 1.19 times ($e^{0.1730}$) significantly more likely to be observed in regions with an increase inequality score (Supplementary Table 8). Furthermore, households are about 0.9 times less likely to remain economically stable when connected to other households in the same *isigodi* ($e^{-0.0907}$), yet 1.34 times more likely to remain stable ($e^{0.297}$) when their network partners also maintain stability (Supplementary Table 9).”

[11] Finally, I did not find any sense in the multinomial 3-way interaction. This may be my mistake, but multinomial regressions are difficult to interpret correctly without adding triple interactions with squared terms making all the assumptions inconceivable. Based on the dependent variable, this should be an ordered logit regression.

[Reply 11] Our choice to use multinomial logistic regression was guided by the proportional odds assumption, which we tested using the Brant test. While we acknowledge the complexity of interpreting multinomial regressions, particularly with 3-way interactions, our analysis showed that the simplification to an ordered logit model was not suitable for our data, as indicated by the Brant test results ($p = 0.0015$) and further supported by the likelihood ratio test comparing multinomial and ordered logit models ($p = 0.0016$). To avoid confusion, we have incorporated these notes into the supplementary materials. Nevertheless, in closing, we would like to thank the Reviewer again for the time and all the engaged comments.

Reviewer #2 (Remarks to the Author):

[1] I thought this was an interesting paper, applying a complexity approach to the enduring question of wealth inequality in South Africa. The methodological approach seems sound and well-justified to me. I think the methodology employed would be very useful to other researchers in the field examining related question, particularly given that there are a number of HDSS sites around South Africa, opening up the possibility for comparative work across and between the sites.

[Reply 1] To begin with, we would like to thank the Reviewer for the time and effort spent reviewing this work. We also acknowledge that our results may have limited generalisability given the unique historical local social dynamics of our study context, but we believe our methodologies may indeed offer an alternative avenue to conceptualise and contextualise family dynamics, particularly in lower-income settings where these family ties may act as a primary safety net. The preceding comments have helped to better position the current study within the broader literature on family dynamics and anthropological works—where our study fits. We especially thank the Reviewer for pointing out many redundancies in our writing.

[2] My main comment is about the presentation of the material, which leaves the reader with some questions which are addressed only later on in the paper about the data, the type of inequality, and some other issues. I would suggest a brief discussion of these earlier on in the piece, possible in the introduction. I would suggest more detail on what is meant by "ownership of resources", the location of the study site, and what the authors understand by the term "emergent". This will help an unfamiliar reader better locate the study.

[Reply 2] We agree that the presentation of the materials, specifically in relation to our data and study objectives, should be more clearly elaborated. As such, we have revised various parts of the manuscript to better explain terminologies when they are first used. For example, we have revised a paragraph in the introduction to more clearly explain the term 'emergent' in our study context:

“Third, several longstanding questions in social and natural sciences relate to the puzzle of whether small-scale social processes can lead to emergent phenomena at the

macro level²⁶⁻³⁰. This micro-to-macro question, pertinent in understanding phenomena such as the recent COVID-19 pandemic³¹ and climate change responses³², seeks to clarify how local social interactions might traverse into population-level dynamics. However, understanding these micro-macro linkages remains a formidable challenge, often due to the methodological complexities involved in harmonising and modelling the expected social patterns and the inherent randomness in human social interactions³³. Previous research has explored the association between macro-level inequality and global network properties^{7,34}, as well as individual-level economic outcomes and local social interactions³⁵⁻³⁷. Yet, the link between macro- and micro-level economic outcomes, alongside the specific types of social interactions that may strengthen returns capable of influencing economic disparities, remains poorly understood. Unpacking the varying network dynamics at play is therefore likely to contribute to a more refined understanding of how to enhance the effectiveness and efficiency of social policy interventions^{31,38}.”

We have also expanded and incorporated sections in the introduction to explain our study population:

“Here, we investigate how, whether, and to what degree the disparities in economic resources may be accentuated by various forms of micro-level social interactions in one of the poorest rural South African settings that has endured repercussions from decade-long racial segregation (apartheid) and more recently, the HIV epidemic. Our study population is located in the uMkhanyakude district in the KwaZulu-Natal province of South Africa, and comprises the Africa Health Research Institute’s (AHRI) Demographic Surveillance Area (DSA)^{39,40}. As a predominantly rural economy historically centred around small-scale farming and animal husbandry, our study context is characteristic of many rural South African settings, with limited institutional support resources and employment opportunities^{39,40}. Access to healthcare, education, and other amenities is hindered by geographic distance and a lack of public transportation.

Demographically, the population is characterised by a high proportion of younger- and middle-aged individuals, with median ages of approximately 22 years for men and 25 years for women in 2018³⁹. Non-residents comprise about 28% of the population, of

whom are often considered as circular labour migrants engaging in temporary movements seeking support, education, and economic opportunities elsewhere, while remaining socio-economically tied to their rural home³⁹. However, the unemployment rate remains steadily high, with nearly 62% of adults lacking formal employment. Consequently, the population relies heavily on non-contributory social grants for household livelihood, particularly the older-age grant, while continuing to be highly mobile^{39,41}.“

[3] Similarly, I think the study could be better located in the South African literature in the introduction. A nod is made to the anthropological lineage into which the study fits, but a brief elucidation of this would be useful.

[Reply 3] Indeed, there exists a long line of anthropological works that have highlighted the importance of these inter-household relationships. We concur with the Reviewer that it will certainly help the reader to better situate our study objectives by acknowledging the existence of prior works. As such, we have revised parts of the introduction to further elaborate the context in which our study fits:

“To construct social networks on a population scale, our work integrates multiple data sources from AHRI DSA, including a population census, a detailed household residency survey, and information on the geographic locations of households. We leverage a unique census indicator—household memberships—to identify inter-household ties formed by overlapping household members for ~12,000 households (~90,000 people) living across an area of over 400km². Here, household membership is defined according to one’s affiliation to a ‘social group’ rather than their actual residency. This definition acknowledges the essential role of absent members in the household economy, where shared identity, obligations, or a common leader foster connections and the flow of resources among social groups⁴¹⁻⁴³. Individuals may therefore maintain multiple concurrent household memberships over time. Such conceptualisation has thus challenged the primacy of viewing households as independent units of analysis, especially among rural South African communities^{14,43-46}.

The significance of these inter-household relationships stems from their role as crucial channels for accessing various forms of resources and support. In lower-income settings, resource pooling primarily occurs informally via social networks given geographic barriers and formal market constraints¹⁷. These resources can be supplied directly through gifts, transfers, or informal loans. They can also be supplied indirectly by disseminating job opportunities and livelihood strategies such as farming¹⁶. These networks, predominantly familial, are often bound by culturally-defined social contracts, compelling families to voluntarily or involuntarily adhere to sharing, caring, and lending practices¹⁴. As such, they may also give rise to uneven economic pressures, with benefits for some translating into losses for others¹⁸. Such reality challenges the presumption that social networks are unequivocally beneficial⁴⁷. While inter-household relationships have been suggested to yield returns capable of influencing a household's economic outcome, however, there has been limited empirical evidence to support this claim. To this end, the question of whether these micro-level social processes have the potential to influence the distribution of economic resources for the communities at large remains underexplored.”

[4] Some aspects of the discussion are opaque. For example, on page 15 the authors state "Nevertheless, observations from this study give us the confidence to hypothesise that network-based efforts to address the negative social consequences under rural poverty will more realistically and effectively achieve the desired outcomes derived from social policy interventions." It is not clear what this means. What are the "network-based effects to address the negative social consequences"? What are the effects, how could they be used to address, and what are the negative social consequences? What sort of social policy interventions are conceived of?

[Reply 4] We thank the Reviewer for highlighting the redundancy in our writing. In closing, we have revised the closing paragraph which now reads as follows:

“There may be other confounding factors that explain our findings, and we cannot prove the cause and effect between social network mechanisms, household resource allocation, and wealth inequality. In this study context, interpersonal relationships and social exchange are rooted in a range of complex historical local social dynamics that we cannot fully evaluate in this work. Nevertheless, observations from this study

highlighted that network-centred strategies—such as those aimed at promoting resource sharing and support systems within social networks—may complement and potentiate the efficacy of social policy intervention designed to reduce disparities in economic opportunities across rural communities.”

Reviewer #3 (Remarks to the Author):

[1] This paper, “Local Network Contagion as a Mechanism for Wealth Inequality,” examines the links between micro-level social networks and macro-level wealth inequality using South Africa’s large survey data. I appreciate the authors’ technical approach to examining the links between micro-level social networks and macro-level wealth inequality with the large-scale empirical dataset.

However, I believe that the authors could improve the paper by addressing the following concerns:

[Reply 1] First, we would like to thank the Reviewer for the time taken to review this work and, foremost, for providing many engaged and constructive suggestions that have helped strengthen the robustness of our findings and reduce redundancies in our writing. Below, we address each comment individually.

[2] First, I suggest that the authors showcase the reproducibility of their analysis in greater detail. The paper’s central argument relies heavily on the results presented in Figure 5, particularly Figure 5i. Therefore, the authors need to demonstrate how robust their findings are, even when using various settings in the statistical model. While the authors touch upon their “robustness check” in Supplementary Note 3, they could conduct additional analyses to show how many errors in the data could be tolerated to arrive at the same conclusion drawn from the analysis. Furthermore, it is essential to test the impact of missing data and the possibility of missing controlled parameters in the statistical results. The authors could show robustness against the missing data or parameters by randomly eliminating a certain percentage of households or parameters from their dataset.

[Reply 2] The key finding arises from our analysis, indeed, is centred around the estimated parameter presented in Fig. 5i. Beyond the commonly applied robustness check for our empirical framework—namely, comparing the observed statistics with a simulated sample from the converged model (GOF test)—we agree with the Reviewer on the importance of assessing the robustness of our findings against alternative, possibly more parsimonious, model specifications. To this end, we ran three additional models with and without various controls to benchmark the estimated parameters against our main results:

Supplementary Table 10. Alternative model specifications with and without household- and regional-level controls.

	MPNet term	Covariate free		Only Regional controls		Only Household-level controls		Main results	
		Parameters	std. error	Parameters	std. error	Parameters	std. error	Parameters	std. error
	TXAX-1A	-0.1548	0.0543	-0.1570	0.0575	-0.1363	0.0570	-0.1354	0.0549
	TXAX-2A	0.2492	0.0848	0.2584	0.0895	0.2358	0.0901	0.2394	0.0838
	L3XAX-1A	0.0870	0.0411	0.0867	0.0397	0.0100	0.0464	0.0094	0.0436
	L3XAX-2A	-0.0200	0.0558	-0.0212	0.0562	-0.0242	0.0601	-0.0230	0.0560
	TXAX-B	-0.0456	0.0251	-0.0226	0.0296	-0.0513	0.0256	-0.0362	0.0333
	L3XAX-1B	0.0032	0.0068	-0.0046	0.0086	0.0047	0.0069	0.0007	0.0096
	L3XAX-2B	0.0025	0.0085	0.0051	0.0092	0.0012	0.0085	0.0012	0.0116
	AllTXAX	0.1005	0.0420	0.0904	0.0425	0.1134	0.0425	0.1024	0.0456

Parameter estimates, derived from the multilevel ALAAMs, are presented as conditional log odds. Descriptions of the estimated parameters can be found in Fig. 4 of the main text. To benchmark these against our main results, we ran three additional models with and without various controls. Findings from these alternative specifications consistently align with our main results.

In addition, we acknowledge that missing data could potentially bias our findings, given that the properties of the entire network might be altered by introducing a small amount of randomness. However, since our study leverages census data covering the entire population, our findings likely retain a degree of representativeness, even considering the missing information. Beyond recognising the potential for selection bias due to missing data patterns, we agree with the Reviewer on the importance of additional robustness checks. Therefore, we have employed the multiple imputation method to impute missing values for all covariates—mostly for the household asset indicator—using a MICE approach in our baseline logistic regression to assess whether our main results differ. The results remain consistent, although we recognise that additional work is necessary to address missingness in social network data (‘Robustness’, Supplementary note 3).

	Imputed model		ALAAM estimates			
	OR	95% CI	OR	95% CI		
Household asset status						
Lowest quintile (ref)						
2 nd lowest	1.3244	1.1826	1.4834	1.3179	1.1787	1.4735
Middle	1.1763	1.0480	1.3204	1.1769	1.0517	1.317
2 nd highest	1.1646	1.0338	1.3119	1.1624	1.033	1.3081
Highest	0.4781	0.4224	0.5411	0.4862	0.4295	0.5504
Household head sex						
Male (ref)						
Female head	1.2027	1.1169	1.2951	1.12	1.0371	1.2096
Mortality experiences						
Zero-death (ref)						
Death	0.8158	0.5531	1.2032	0.7945	0.5404	1.1683
Pension-eligibility						
No (ref)						
Pension	1.3429	1.2374	1.4574	1.2224	1.1247	1.3285
Household size	1.0547	1.0377	1.0721	1.0351	1.0179	1.0526
N of non-residents	1.4261	1.3550	1.5008	1.2619	1.1971	1.3303
N of non-residents ²	0.9723	0.9686	0.9759	0.9815	0.9778	0.9851

[3] Second, I suggest that the authors explain the statistical model and the interpretation of results more clearly. The multilevel network analysis shows whether a “network configuration occurs more than we expect by chance, conditional on the rest of the estimated parameters in the model,” and all the results from the model indicate associations and not causalities. Although the authors have clarified this point in the manuscript, some interpretations suggest causal relationships. For instance, in the abstract, the authors use “support,” “promote,” and “contribute to an increase” to describe their findings, but such terms imply causal directionality, which is misleading for their association analysis. If the authors can justify their causal claims from their analysis, they should elaborate more on

them. Otherwise, the authors should avoid using terms that imply causal directionality over the paper and instead focus on the associations revealed by their analysis.

[Reply 3] We thank the Reviewer for highlighting this point. Indeed, our approach does not fully address endogeneity concerns nor does it establish causality. We have thus revised many parts of the manuscript to avoid making causal claims. Specifically, the abstract has been updated to read as follows:

“Given limited institutional resources, low-income populations often rely on social networks to improve their socioeconomic outcomes. However, it remains in question whether small-scale social interactions could affect large-scale economic inequalities in under-resourced contexts. Here, we leverage population-level data from one of the poorest South African settings to construct a large-scale, geographically defined, inter-household social network. Using a multilevel network model, we show that having social ties in close geographic proximity is associated with stable household asset conditions, while distant ties correlate to changes in asset allocation. Notably, we find that localised network contagion effects are associated with an increase in wealth inequality at the regional level, demonstrating how emergent inequality may arise from micro-level social processes. Our findings highlight the importance of understanding complex social connections underpinning inter-household resource dynamics, and raise the potential of large-scale social assistance programs to reduce disparities in resource-ownership by accounting for local social constraints.”

[4] Related to this, I suggest that the authors clarify the statistical model even in the main text. In my first reading, I did not clearly understand that each network composition shown in Figure 5 is nested in the analysis, as shown in Figure 7. My confusion came from what the light-colored (circle and square) nodes indicate in Figure 5. Now I understand the light-colored nodes mean “unspecified”; i.e., for example, Fig. 5i’s composition is nested in Fig 5d’s one, and Fig. 5d’s one is nested in Fig 5a’s one. However, in Figure 4, the authors seem to use the light-colored nodes for “no change” in opposition to the dark-colored nodes; in that case, Fig 5i’s composition should complement Figure 5d’s. The authors should clarify this point to avoid possible confusion.

[Reply 4] This is indeed a redundancy when communicating and visualising our findings, and we thank the Reviewer for highlighting this. We have incorporated additional stylised figures to more clearly illustrate ALAAM parameters in the legends of Fig. 4 and Fig. 5.

[5] Third, I recommend that the authors provide more nuanced explanations of the association link to clarify the study's importance. Community-level wealth inequality is calculated based on the accumulation of individual wealth. Thus, when the households in a community have no changes in wealth, the community's wealth inequality has no way to increase or decrease for calculation. The main result shown in Fig. 5i ostensibly confirms the apparent micro-macro association. I think their analysis should provide more nuanced explanations, but they are unclear. For example, does Figure 5i's composition still hold the significant odd even controlling the composition where wealth-mobilized households individually belong to inequality-increased communities without connections between them? If so, the network effect would be highlighted. Furthermore, from the perspective of addressing the societal problem, it could be more important to show how network mechanisms help people reduce wealth inequality rather than increase it. Thus, It would be beneficial to investigate whether the result would be just the opposite of Figure 5 if the community-level binary variable is set in the opposite way.

[Reply 5] We thank the Reviewer for this engaged comment that has helped to highlight the network effect in our model. Our main model does include a parameter that indicates the independent effects of wealth-mobilised households on regional-level inequality outcome (Cross-level contagion, Supplementary Table 5). This parameter indicates whether household- and regional-level outcomes are interdependent, and is not statistically significant. As such, we have elaborated this in the manuscript to highlight the network effect as "...This [(Fig. 5i)] parameter remains significant after adjusting for the independent effects of wealth-mobilised households in regions with rising inequality but without mutual connections (Supplementary Table 5)."

Moreover, our result is just the opposite—particularly, Figure 5i—when we set the regional-level binary outcome in the opposite direction (0 = increase/stable inequality score; 1 = decrease score). Specifically, the estimated parameter suggests that regional-level inequality score is negatively associated with local network contagion processes: A decrease in the inequality score is about 0.902 times less likely to be observed in regions where households are socially connected and economically contagious. This may suggest

that network effects are closely linked to the dynamics of wealth inequality. However, this statement is not directly tested in our study, and we consider it an important area for future research to explore more definitively.

[6] Finally, I would suggest that the authors describe the people who connect the households (or “social groups”) in more detail, e.g., by giving examples and showing the percentages in a table. The interpretation of a network analysis depends on the substance of the network links (or edges), and the network effect could differ according to the link types. Thus, such additional analyses would help readers imagine how the bridging people work on economic mobilization in the specific field and context. They would also help consider how this study’s findings could be applied to other populations.

[Reply 6] As the Reviewer rightly pointed out, detailing under what circumstances two households would nominate the same member(s) is important, as, in our study context, they showcase how wealth and assets may be distributed across social groups. In Supplementary Note 1 and Supplementary Fig. 1, we have examined who were more likely to have concurrent household memberships. We have also referred back to prior studies in this population that have detailed the nature of these inter-household relationships (as shown below). Nevertheless, as our focus is not entirely on the formation of the inter-household social network, we believe that a more thorough examination of the structural dynamics of this network would be more appropriate—potentially through the use of social selection models (e.g., p^* model).

“Our study population comprises 11,834 households (92,688 people) observed in 2016, scattered across 23 *izigodi*³⁹. To construct the social network for this population, we integrate a prospective, detailed household residency survey spanning from 2000 to 2016 to identify inter-household ties based on overlapping household members. Here, we consider two households as connected if they have shared a common member in the study timeframe (Fig. 1a). AHRI DSA distinguishes one’s household memberships (household) with their actual residency status (homestead) to capture the ‘stretched households’ arrangements observed across many rural South African settings, whereby families work together socio-economically while living apart⁴³. In this population, a concurrent membership reflects inter-household ties that are predominantly driven by a combination of shared responsibilities (care receiving and giving), authority (headship

and polygamy), and identity (kinship) between households, as well as by historical social relations that may give rise to a sense of belongingness between families^{41,43,45}. In these instances, household members are thus usually, but not always, related. “

Minor comments:

[7] In Figure 4, the authors colored the areas where the rising inequality scores are more than 0 but mentioned the areas where they are more than 0.02 in the caption.

[Reply 7] We set this threshold to ensure that the increase in inequality score is not driven by demographic changes, such as changes in population size in this highly mobile population (Supplementary Table 2). We therefore limited our analysis to regions ($n = 6$) where the increase in inequality score is above the 75th percentile, equivalent to an increase exceeding 2% ($\Delta Inq_v \geq .02$).

[8] Figure 2a lacks the labeling of the light-colored part.

[Reply 8] We believe this may be due to a formatting error on our end, and we thank the reviewer for highlighting it.

[9] The authors use “isigodi” and “izigodi” for the same meaning, including the figures. Please unify the terms.

[Reply 9] *Isigodi* refers to a single region, while *izigodi* refers to multiple regions. In closing, we again thank the Reviewer for the time and all the critical and constructive comments.

Reviewers' Comments:

Reviewer #1:

Remarks to the Author:

I extend my gratitude to the editor for granting me the opportunity to review this manuscript once again, and I am thankful to the authors for their comprehensive responses to my inquiries and comments from the previous round of review. I believe the manuscript has undergone significant improvements and clarifications. However, even in its current form, there remain several weaknesses within the study that require resolution.

I appreciate the authors' additions and modifications to the text and concur with the notion that network analyses provide a valuable bridge between micro-level decision-making and macro-level structural outcomes. Nonetheless, I feel that the network perspective has not been sufficiently developed within the study. Based on my suggestion, the authors calculated a number of network science metrics that enhanced the data description. However, my original feedback was not merely a request for the network's average path length or clustering coefficient; rather, it was aimed at understanding how the properties of the network under investigation relate to those of other typical network structures observed in reality, such as random, scale-free, or small-world networks. While the calculation and tabulation of network descriptors aid in understanding the network, the taxonomic listing without context primarily serves to fulfil my specific requests, whereas the goal should be to understand the unique aspects of this network.

Similarly, my feelings extend to the visualization of networks. Although viewing the network brought us closer to understanding it, this knowledge is not leveraged elsewhere in the study. The inclusion of random-walk networks, which the authors do not reference or connect to any questions or descriptions, is perplexing and appears irrelevant.

Moreover, there is further confusion regarding the econometric calculations. The authors' response indicated the necessity for multinomial logistic regressions, yet I was unable to find results for such regressions either in the results tables or the Supplementary Notes (SN).

Also, in SN "Note 1. Data", the second equation describes this regression, including a three-way interaction term, one of which is a quadratic term. I invite the authors to interpret the coefficient of this variable for me, to understand its significance.

Furthermore, the authors have yet to justify why classifying the direction of change in the dependent variable as either negative or positive is a correct identification strategy. Currently, if $w_h,2018 > w_h,2016$, it is classified as 1, and if $w_h,2018 < w_h,2016$, it is also classified as 1. The only condition for the logistic regression's dependent variable to be 0 is if $w_h,2018 = w_h,2016$. Why do the authors ignore the direction of change in w_h or inq_v ?

In summary, while I recognize the authors' effort and the sincerity of their responses, numerous improvements are still needed in the accuracy of network science and statistical analyses and/or their proper communication.

Reviewer #2:

Remarks to the Author:

Thank you to the authors for their responses to my queries. I am satisfied with the responses and amendments.

A few small comments:

Line 42: what are "offline behaviors"?

On page 1 and 2, it seems equivalence is drawn between rural and poor, but this is not always the case. Perhaps some care is warranted here not to elide the two.

Line 94: I believe the correct name is the older person's grant.

Reviewer #3:

Remarks to the Author:

I appreciate that the authors have taken my review into careful consideration. Their revised version has addressed many of my concerns, particularly regarding the analysis's robustness (the first) and context specification (the fourth). It would be beneficial if the authors mention the robustness of the results from multiple aspects in the main text, in addition to the supplementary section, because it helps understand the likelihood of the network effects.

The authors have also addressed my other concerns, which I am grateful for. However, I still see some room for further clarification regarding the confusing causal claims (the second) and the nuanced explanation of the main result (the third). In addition, I suggest that the authors clarify this research contribution in both the introduction and discussion sections.

As the authors agree, their analysis does not directly examine causalities. They have revised the abstract to better represent their analyses in response to my concern. However, some of their claims can still create confusion about the causality. For example, they have used the term "network contagion" in the title and the main text, which implies that something is transmitted from Actor A to Actor B. However, the authors admit that their analysis does not differentiate between the contagion effect and the social-selection effect (on page 10). Although the authors excuse the broad definition of "network contagion" to include the possibility of social selection, the term has been carefully used to distinguish it from social selection (e.g., Christakis, N. A. & Fowler, J. H. The spread of obesity in a large social network over 32 years. *N. Engl. J. Med.* 357, 370–379 (2007)). Since this study does not differentiate between the contagion effect and social selection, the authors should avoid using the term "contagion" to describe their findings. If there are any references that support the author's claim of "network contagion," they should be cited.

I am also concerned about the authors' usage of the term "emergence" to describe the primary finding of Fig. 5i. In this context, "emergences" refer to micro-social interactions that result in social outcomes, such as regional-level inequality. However, it is possible that regional-level inequality also leads to specific social connections and tie formations at the micro level, which we do not categorize as "emergence." I am unsure how this study's analysis can exclude the reverse causality. I recommend that the authors clarify this point. If the result of Fig. 5i includes both causal and reverse causal relationships, the authors should avoid using the term "emergence."

I appreciate the authors' further analyses to give more nuance about the main result of Fig. 5i, as well as the additional text pointing to Supplementary Table 5. However, I found it more helpful if the authors could elaborate further on what the additional analyses suggest about the main finding of Fig. 5i after the additional text. I also suggest that the authors discuss more about the theoretical prediction and importance related to Fig. 5i's findings. The introduction discusses possible contributions of distant workers in this study's focal context, eventually shown in Fig. 5c. Similarly, further introduction about the Fig. 5i effect could clarify its importance. Again, if the authors claim the "emergence" of a network effect then, I suggest they discuss the possibility of reverse causality."

Finally, I would be grateful if the author could provide more concrete explanations in the introduction and discussion sections. There are two areas that I think need improvement. Firstly, the data

collection method needs clarification. In the introduction, the authors mention that their data-collection method is a unique approach to gathering information from rural and low-income populations at scale. They suggest that survey-based approaches are insufficient in this regard. However, it is unclear how their method differs from a typical national census, which is also a type of survey. I would appreciate it if the authors could clarify their advancement in data collection and how it differs from previous census studies. Secondly, the policy implementation discussion section could be more specific. While I agree that this study's population-level multi-layer analysis can benefit policymaking, the current discussion on possible policy implementation is general and vague. Based on their findings, the authors should suggest specific policies that could help mitigate wealth inequality in this region. This would help readers understand the significance of this study's findings and make it more applicable in real-life situations.

Minor comments:

- Thank you for bringing to my attention the relation between the terms "isigodi" and "izigodi." As a typical reader, I was not aware that "isigodi" is the singular form of "izigodi." Therefore, it would be helpful if you could specify "isigodi" when you first introduce "izigodi" in the main text.

- I noticed that the authors use many research questions that begin with "whether." However, I believe that "how" questions may be more appropriate, as the authors don't answer these questions with a simple yes or no.

- I appreciate that the authors have provided descriptive statistics in Figures 2 and 3. However, I am unsure of how these statistics and figures are related to the main network analysis. It would be helpful if the authors could explain the meaning of the descriptive statistics in relation to the main analysis.

- I was confused by the term "density" on the y-axis of Fig. 1b because network science uses "density" in a different sense in this figure. The authors should specify the term ("Degree" on the x-axis could confuse readers outside of network science).

- I did not understand what the network diagrams of the random walk represent in Fig 1c. Please specify them in the main text (or I would recommend removing them).

Reviewer #1 (Remarks to the Author):

[1] I extend my gratitude to the editor for granting me the opportunity to review this manuscript once again, and I am thankful to the authors for their comprehensive responses to my inquiries and comments from the previous round of review. I believe the manuscript has undergone significant improvements and clarifications. However, even in its current form, there remain several weaknesses within the study that require resolution.

[Reply 1] To begin, we would like to thank the Reviewer for taking the time to provide another set of constructive evaluations and comments on this work, particularly for pointing out many potential directions to further develop a network perspective into this study. We agree with the Reviewer that there are aspects of how we communicate our findings and methods that need clearer elaboration and justification. This revision has therefore incorporated additional supplementary analyses to better understand the characteristics of the constructed network, along with added text to elaborate on these analyses in detail; we have also clarified the econometric calculations and our choice of outcome variable classification. Below, we address each comment individually.

[2] I appreciate the authors' additions and modifications to the text and concur with the notion that network analyses provide a valuable bridge between micro-level decision-making and macro-level structural outcomes. Nonetheless, I feel that the network perspective has not been sufficiently developed within the study. Based on my suggestion, the authors calculated a number of network science metrics that enhanced the data description. However, my original feedback was not merely a request for the network's average path length or clustering coefficient; rather, it was aimed at understanding how the properties of the network under investigation relate to those of other typical network structures observed in reality, such as random, scale-free, or small-world networks. While the calculation and tabulation of network descriptors aid in understanding the network, the taxonomic listing without context primarily serves to fulfil my specific requests, whereas the goal should be to understand the unique aspects of this network.

[Reply 2] We thank the reviewer for this follow-up comment. Given the distinctive nature of our network construction approach, we agree that the manuscript would benefit from a more elaborated description to contextualise our observed network, particularly in relation to those topologies commonly observed in social network research. We do wish to emphasise that our initial reluctance to elaborate stemmed from the fact that our empirical framework aims to model the process of social influence (ALAAM) where the network structure is considered exogenous and the effects in ALAAM assess how the given network structure impact the micro and macro level nodal outcomes. Investigating the structural properties of the network is a different research question and may be better addressed by Exponential Random Graph Models (ERGM) for social selection processes. We do agree, however, that investigating the network structure can help better understanding of the system and assist in interpretation. To address this comment more thoroughly, we have included additional text, a table, and a figure in the Results section and/or Supplementary Note 1 to further elucidate this aspect. In the main text. In the main text, it can be read as follows:

“Descriptively, this sparsely connected undirected network consists of 10,162 inter-household ties and follows a long-tailed degree distribution with an average degree of 1.72 (SD = 1.8) per household (Fig. 1b). This network has a global clustering coefficient of approximately 0.21 and an average path length of roughly 11.84 (SD = 2.98). Over 80% of households, approximately 8,453 in total, are connected through a shared member. The largest connected component in this network consists of about 6,462 households (Fig. 1c). We compare this observed network with theoretical models within the broader landscape of network topologies, such as the random⁵¹, small-world³⁰, and scale-free²⁶ types (Supplementary Table 2). Compared to a random network, which has a lower clustering coefficient and a shorter average path length, our network shows a degree of clustering indicative of a more structured, possibly hierarchical arrangement. Although the average path length and local clustering are closer to that observed in small-world networks, the degree distribution does not align with the uniformity seen in such models, nor does it fit the hub-and-spoke configuration typical of scale-free networks. These observations suggest that our observed network may exhibit community structure with a quasi-small-world configuration, where a small share of ties could play a key role in enhancing overall connectivity and functionality³⁰.

To further explore the configurations of the constructed network, we delineate the local substructures of the largest connected component using random walks (Fig. 1c). In addition to the typical spanning-tree-like structure found in family lineage networks⁴⁵ (Random walk 3), these substructures uncover diverse embeddedness patterns that have not been widely reported in studies concerning the utility and economic functions of informal family support systems across lower-income settings¹³⁻¹⁷. We discuss the setting, data conceptualisation, and network construction method in greater detail in the Methods section and Supplementary Note 1.”

[3] Similarly, my feelings extend to the visualization of networks. Although viewing the network brought us closer to understanding it, this knowledge is not leveraged elsewhere in the study. The inclusion of random-walk networks, which the authors do not reference or connect to any questions or descriptions, is perplexing and appears irrelevant.

[Reply 3] Presenting these visualisations without context was indeed a limitation in the previous revision. As presented above, the purpose of these figures is to deconstruct the global network visualisation by illustrating smaller, more locally defined social interactions. We then relate these visuals to both observational and empirical works within and beyond our study population to highlight a lesser-acknowledged aspect of kin-based social interactions: the structural arrangement of these networks. This is done to emphasise on the potential implications of our data construction method for understanding network structural effects on social and health outcomes within and beyond our study populations.

[4] Moreover, there is further confusion regarding the econometric calculations. The authors' response indicated the necessity for multinomial logistic regressions, yet I was unable to find results for such regressions either in the results tables or the Supplementary Notes (SN). Also, in SN "Note 1. Data", the second equation describes this regression, including a three-way interaction term, one of which is a quadratic term. I invite the authors to interpret the coefficient of this variable for me, to understand its significance.

[Reply 4] We thank the Reviewer for pointing out this confusion, and we must acknowledge an error on our end in communicating the interaction term in the original equation. This supplementary regression included a pair of two-way interactions to understand how household membership statuses vary by age, sex, and marital status. Building on prior work in the same context, we expected gendered and age-related patterns in the number of household memberships. We hypothesized that women might receive the highest number of membership nominations while in informal unions but prior to formal marriage, and that men might receive more nominations at older ages, potentially due to headship and/or marital practices. The equation has now been revised as follows:

$$\log \left(\frac{P(Y = j)}{P(Y = J)} \right) = \alpha + \beta_1 \text{sex} + \beta_2 \text{age} + \beta_3 \text{age}^2 + \beta_4 \text{marital} + \beta_5 (\text{sex} \times \text{age}) + \beta_6 (\text{sex} \times \text{age}^2) + \epsilon_j$$

The first interaction term suggests that women have an increasing likelihood of receiving multiple membership nominations as they age (RRR = 1.0678). However, this increase in likelihood diminishes as they age, as indicated by the quadratic term interaction (RRR = 0.999). These interactions imply that while age is generally a positive factor for women in receiving nominations, its effect is less pronounced as women reach higher age brackets, possibly due to changing marital status and/or social roles.

We had included figures visualizing the marginal effects of age on the number of household membership reported, by sex, and of marital status on the outcome variable in Supplementary Figure 1. To avoid further confusion, we have added a table detailing the regression coefficients in Supplementary Table 1.

Supplementary Table 1. Multinomial logistic regression predicting the relative risk ratio (RRR) of having one (base), two, or three-plus household memberships among individuals in 2016, by age, sex, and marital status.

	Two household memberships				Three or more household memberships					
	RRR	std. error	$P > t $	95% CI	RRR	std. error	$P > t $	95% CI		
Sex										
Male (base)										
Female	0.4575	0.0925	0.0000	0.3078	0.6798	0.2204	0.2359	0.1580	0.0270	1.7962
Age	0.9251	0.0073	0.0000	0.9109	0.9396	0.9311	0.0253	0.0090	0.8827	0.9821
Sex * Age										
Female	1.0678	0.0114	0.0000	1.0457	1.0904	1.1495	0.0710	0.0240	1.0183	1.2974

Age ²	1.0009	0.0001	0.0000	1.0007	1.0010	1.0010	0.0003	0.0000	1.0005	1.0015
Sex * Age ²										
Female	0.9990	0.0001	0.0000	0.9988	0.9993	0.9975	0.0008	0.0030	0.9959	0.9992
Marital status										
Never-married (base)										
Married	0.9077	0.0773	0.2550	0.7682	1.0725	0.9382	0.3480	0.8640	0.4535	1.9412
Widowed	0.5851	0.0832	0.0000	0.4428	0.7731	0.7400	0.4570	0.6260	0.2206	2.4829
Separated/divorced	1.0704	0.4202	0.8630	0.4959	2.3105	3.2216	3.3729	0.2640	0.4139	25.0747
Informal union	1.1016	0.0554	0.0540	0.9983	1.2156	0.8079	0.1917	0.3690	0.5074	1.2864
Constant	0.2438	0.0351	0.0000	0.1839	0.3232	0.0087	0.0048	0.0000	0.0029	0.0256

[5] Furthermore, the authors have yet to justify why classifying the direction of change in the dependent variable as either negative or positive is a correct identification strategy. Currently, if $w_{h,2018} > w_{h,2016}$, it is classified as 1, and if $w_{h,2018} < w_{h,2016}$, it is also classified as 1. The only condition for the logistic regression's dependent variable to be 0 is if $w_{h,2018} = w_{h,2016}$. Why do the authors ignore the direction of change in w_h or inq_v ?

[Reply 5] To examine the association between changes in regional- and household-level outcomes, we initially hypothesize that an increase in regional-level inequality could be related to a combination of upward, downward, and stable asset movements. As such, our primary model classifies the outcome variable as any change. Nonetheless, we have tested various combinations of outcome changes and decided to include these analyses in the supplementary material (Supplementary Table 8-10), mainly because our model does not estimate all combinations within a single structural framework. We determined that focusing exclusively on a single estimate in the main text—whether of upward, downward, or stable asset status—might not provide an entirely accurate interpretation of their relationships with regional-level wealth variations. In other words, and as shown in a series of supplementary analyses, inequality may arise given a combination of upward, downward, and stable asset movements throughout the observation period. Our identification strategy, therefore, aims exclusively to test for the existence of micro-macro linkages and, if present, to determine whether the proximity of these social ties matters. We nevertheless agree that additional

clarity in writing would be beneficial, and we have included text to document these reasons as follows:

“At the household level (denoted as h), we create a binary outcome variable to measure if there has been any change in a household’s relative wealth quintile between the two timeframe ($\mathbf{1}_{\{\Delta w_h \neq 0\}}$ where $\Delta w_h = w_{h,2018} - w_{h,2016}$). To understand the disparity in asset wealth at the regional level, we utilise the household wealth index to construct a measure of wealth inequality for each *isigodi* (denoted as v), following a similar approach to constructing a Gini index (Supplementary Note 2). We then create a binary variable to identify regions that have experienced an increase in their inequality score over time ($\mathbf{1}_{\{\Delta Inq_v > 0\}}$ where $\Delta Inq_v = Inq_{v,2018} - Inq_{v,2016}$). The household-level binary classification is chosen to identify changes in wealth within households rather than the direction of these changes, since an increase in regional wealth inequality could result from various combinations of upward, downward, and stable asset movements. Given the constraints of our empirical framework (see ‘Statistical analyses’ in Methods), we conduct three sensitivity analyses where the binary outcome is defined as: i) upward; ii) downward; and iii) no change in asset quintile (see Supplementary Note 3).

[6] In summary, while I recognize the authors' effort and the sincerity of their responses, numerous improvements are still needed in the accuracy of network science and statistical analyses and/or their proper communication.

[Reply 6] In closing, we wish to thank the Reviewer again for the time for providing another round of engaged and constructive comments.

Reviewer #2 (Remarks to the Author):

[1] Thank you to the authors for their responses to my queries. I am satisfied with the responses and amendments.

[Reply 1] To begin, we would like to extend our gratitude to the Reviewer for taking the time to provide another round of careful and constructive reviews.

A few small comments:

[2] Line 42: what are "offline behaviors"?

[Reply 2] We refer to offline behaviours as those occurring beyond interactions on digital platforms, but we agree the term can be more clearly defined as “social interactions”. The text now reads as follows:

“However, it remains in question whether these digital proxies can capture more tangible, costly, and culturally-defined social interactions related to resource-sharing and exchange activities among families in poverty-stricken settings^{17,18}”.

[3] On page 1 and 2, it seems equivalence is drawn between rural and poor, but this is not always the case. Perhaps some care is warranted here not to elide the two.

[Reply 3] This is an important point. Although our study context is in one of the more economically deprived rural settings, we agree that care is needed to avoid equating the two terms. We have edited the text to address this concern as follows:

“First, limited studies have been conducted in low-income populations^{2,4}. Previous research has highlighted the importance of social networks as critical support systems in poorer environments¹³⁻¹⁵. Yet, empirical investigation exploring the influence of social networks on inequalities in these settings pose a significant scalability challenge. This

limitation may be hampered by high costs and inadequate infrastructure for extensive research activities, particularly among those located in remote settings. As a result, existing research tend to focus on specific population sub-groups¹⁶, lacking data on network interactions observed at the population level. In response, computational and online experimental techniques may offer scalable alternatives^{7,9,11}. However, it remains in question whether these digital proxies can capture more tangible, costly, and culturally-defined social interactions related to resource-sharing and exchange activities among families in poverty-stricken settings^{17,18}. The availability of population-level in-person data therefore offers an important opportunity to gain a deeper understanding of the under-researched correlation between social networks and economic inequality in rural or poorer contexts.”

[4] Line 94: I believe the correct name is the older person's grant.

[Reply 4] We thank the Reviewer for highlighting this point. We have revised the manuscript to more accurately describe the subject as either “older person’s grant” or “old age pension”. In closing, we appreciate the Reviewer’s time dedicated to another round of reviews.

Reviewer #3 (Remarks to the Author):

[1] I appreciate that the authors have taken my review into careful consideration. Their revised version has addressed many of my concerns, particularly regarding the analysis's robustness (the first) and context specification (the fourth). It would be beneficial if the authors mention the robustness of the results from multiple aspects in the main text, in addition to the supplementary section, because it helps understand the likelihood of the network effects.

[Reply 1] To begin, we wish to thank the Reviewer once again for their time and for another round of engaged, constructive, and careful reviews. We agree with the Reviewer that further clarity in writing is required to better highlight the strengths and weaknesses of this work, and we especially thank the Reviewer for carefully highlighting these aspects throughout the review process. In this revision, we have further refined the manuscript to avoid causal interpretations, and have revised sections of the introduction and results to clarify the strengths and weaknesses of this work. Below, we address each comment individually

[2] The authors have also addressed my other concerns, which I am grateful for. However, I still see some room for further clarification regarding the confusing causal claims (the second) and the nuanced explanation of the main result (the third). In addition, I suggest that the authors clarify this research contribution in both the introduction and discussion sections.

[Reply 2] It is indeed important to clarify that our methodological approach does not fully address endogeneity nor establish causality. Accordingly, we have revised several parts of the manuscript to address this concern, particularly in the Results, Methodology, and Figures sections. We have elaborated on this point in greater detail in the following responses.

[3] As the authors agree, their analysis does not directly examine causalities. They have revised the abstract to better represent their analyses in response to my concern. However, some of their claims can still create confusion about the causality. For example, they have used the term "network contagion" in the title and the main text, which implies that something is transmitted from Actor A to Actor B. However, the authors admit that their analysis does not differentiate

between the contagion effect and the social-selection effect (on page 10). Although the authors excuse the broad definition of "network contagion" to include the possibility of social selection, the term has been carefully used to distinguish it from social selection (e.g., Christakis, N. A. & Fowler, J. H. The spread of obesity in a large social network over 32 years. *N. Engl. J. Med.* 357, 370–379 (2007)). Since this study does not differentiate between the contagion effect and social selection, the authors should avoid using the term "contagion" to describe their findings. If there are any references that support the author's claim of "network contagion," they should be cited.

[Reply 3] The terminology used has been carefully revised to avoid implying causality. While “contagion” is a theoretical concept we aim to explore empirically, we acknowledge the potential for misinterpretation as a causal claim. Upon consideration, we have opted for the term “interaction”, which may avoid such conflation. The entire manuscript has thus been updated to replace contagion with *interaction*, with the most significant change being the title, now reading “Local Network Interaction as a Mechanism of Wealth Inequality”

[4] I am also concerned about the authors' usage of the term "emergence" to describe the primary finding of Fig. 5i. In this context, "emergences" refer to micro-social interactions that result in social outcomes, such as regional-level inequality. However, it is possible that regional-level inequality also leads to specific social connections and tie formations at the micro level, which we do not categorize as "emergence." I am unsure how this study's analysis can exclude the reverse causality. I recommend that the authors clarify this point. If the result of Fig. 5i includes both causal and reverse causal relationships, the authors should avoid using the term “emergence.”

[Reply 4] We again thank the Reviewer for this engaged comment. While the concept of emergence is explicitly a theoretical objective that we wish to examine, however, to address the stated concern more thoroughly, we instead propose a more direct terminology by stating it as “micro-macro link”. This correction can be found specifically in Fig. 5 and Fig. 7. Yet, we have retained the term “emergence” *only* when we are referring it as a theoretical concept, separately from discussing our empirical findings.

[5] I appreciate the authors' further analyses to give more nuance about the main result of Fig. 5i, as well as the additional text pointing to Supplementary Table 5. However, I found it more helpful if the authors could elaborate further on what the additional analyses suggest about the main finding of Fig. 5i after the additional text. I also suggest that the authors discuss more about the theoretical prediction and importance related to Fig. 5i's findings. The introduction discusses possible contributions of distant workers in this study's focal context, eventually shown in Fig. 5c. Similarly, further introduction about the Fig. 5i effect could clarify its importance. Again, if the authors claim the "emergence" of a network effect then, I suggest they discuss the possibility of reverse causality.

[Reply 5] Given the complex local social dynamics of our study population, we aimed to examine how connections spanning diverse regions relate to both household- and regional-level economic outcomes. We initially elaborated on these theoretical predictions in the section titled "Modelling Multilevel Networked Outcomes". However, we agree that a follow-up description would certainly offer a better clarification on the importance of this "micro-macro link" parameter. As such, we have revised parts of the Result section to more fully address this suggestion as follows:

"Our model revealed that an increase in the regional-level inequality score is positively associated with local network interaction processes (Fig. 5i). Specifically, a rise in the inequality score is about 1.11 times more likely to be observed in regions where households are socially connected and economically interdependent. This parameter remains significant after adjusting for the effects of wealth-mobilised households in regions with rising inequality but without mutual connections – a parameter that is not statistically significant ('Cross-level interaction' in Fig. 7 and Supplementary Table 6). The robustness of this finding is further supported by alternative, more parsimonious model specifications (Supplementary Table 11). These analyses suggest that while social connections may span across multiple regions – notably given the high level of temporary residents of our study population – co-located social ties appear to be a major source of both insurance and influence on the patterns and dynamic of overall wealth variations.

Nevertheless, we recognise the possibility of reverse causality, where regional inequality patterns may also influence the formation and strength of local social interactions. To partly mitigate concerns that changes in regional outcomes are due to other unobserved factors, such as changing population size, we compare the demographic composition across all regions and confirm a consistent population size (Supplementary Table 3). In this conclusion, our analyses may capture some degree of the network interaction effects associated with increased regional inequality against a backdrop of a stable population and after controlling for individual attributes, network dependencies, and structural effects.”

[6] Finally, I would be grateful if the author could provide more concrete explanations in the introduction and discussion sections. There are two areas that I think need improvement. Firstly, the data collection method needs clarification. In the introduction, the authors mention that their data-collection method is a unique approach to gathering information from rural and low-income populations at scale. They suggest that survey-based approaches are insufficient in this regard. However, it is unclear how their method differs from a typical national census, which is also a type of survey. I would appreciate it if the authors could clarify their advancement in data collection and how it differs from previous census studies. Secondly, the policy implementation discussion section could be more specific. While I agree that this study's population-level multi-layer analysis can benefit policymaking, the current discussion on possible policy implementation is general and vague. Based on their findings, the authors should suggest specific policies that could help mitigate wealth inequality in this region. This would help readers understand the significance of this study's findings and make it more applicable in real-life situations.

[Reply 6] The initial purpose of referencing a survey-based approach was to emphasise that prior research on social network effects in developing economies tends to be smaller in scale, focusing mainly on sub-population, and with limited evidence concerning the role of network structural factors. However, we believe this confusion stems from our end in communicating and motivating the contribution of our study — and we thank the Reviewer for pointing out this redundancy in writing. The text is now revised as follows:

“First, limited studies have been conducted in low-income populations^{2,4}. Previous research has highlighted the importance of social networks as critical support systems in poorer environments¹³⁻¹⁵. Yet, empirical investigation exploring the influence of social networks on inequalities in these settings pose a significant scalability challenge. This limitation may be hampered by high costs and inadequate infrastructure for extensive research activities, particularly among those located in remote settings. As a result, existing research tend to focus on specific population sub-groups¹⁶, lacking data on network interactions observed at the population level. In response, computational and online experimental techniques may offer scalable alternatives^{7,9,11}. However, it remains in question whether these digital proxies can capture more tangible, costly, and culturally-defined social interactions related to resource-sharing and exchange activities among families in poverty-stricken settings^{17,18}. The availability of population-level in-person data therefore offers an important opportunity to gain a deeper understanding of the under-researched correlation between social networks and economic inequality in rural or poorer contexts.”

For the policy recommendation, we agree with the Reviewer that it is important to contextualise these findings to inform more concrete policy formulation. As our study does not directly address policy-relevant parameters, however, we thought a way forward may be to highlight a policy-related parameter that we did model (older person’s grant) and to underscore its policy relevance in the discussion – or to consider/motivate how to integrate a network perspective to improve policy effectiveness in this study context – for future research. We concur with the Reviewer in this regard, though we think this is beyond the scope of this study.

Minor comments:

[7] Thank you for bringing to my attention the relation between the terms “isigodi” and “izigodi.” As a typical reader, I was not aware that “isigodi” is the singular form of “izigodi.” Therefore, it would be helpful if you could specify “isigodi” when you first introduce “izigodi” in the main text.

[Reply 7] We have now included an additional sentence to clarify these terminologies as follows:

“We then extend this asset-based index to the regional level, summarising the patterns and dynamics of wealth inequality for each of the 23 administrative units within the DSA (singular *isigodi*, plural *izigodi*).”

[8] I noticed that the authors use many research questions that begin with “whether.” However, I believe that “how” questions may be more appropriate, as the authors don’t answer these questions with a simple yes or no.

[Reply 8] We thank the Reviewer for this engaged comment. Where possible, we have now more carefully revised the stated question to better align with our empirical objective, and one example would be:

“we investigate how, and to what degree, the disparities in economic resources may be accentuated by various forms of micro-level social interactions in one of the poorest rural South African settings that has endured repercussions from decade-long racial segregation (apartheid) and more recently, the HIV epidemic.”

Another relevant example where we are indicating a yes/no question is as follows:

“Here, we estimate whether a change in a household’s economic conditions is associated with corresponding changes in their network peers. “

[9] I appreciate that the authors have provided descriptive statistics in Figures 2 and 3. However, I am unsure of how these statistics and figures are related to the main network analysis. It would be helpful if the authors could explain the meaning of the descriptive statistics in relation to the main analysis.

[Reply 9] We decide to include these analyses to emphasise context-specific parameters that are well-documented in relation to health outcomes across demographic surveillance sites in South Africa, specifically the role of old age pensions and the sex of household heads. This

also serves to highlight that our statistical framework incorporates a series of household- and regional-level controls. Descriptions for these parameters are provided as follows:

“Our analysis begins by estimating two baseline logistic regressions predicting outcomes observed at both levels, without accounting for network parameters. We first introduce a range of covariates for both models (see ‘Statistical analysis’ in Methods). These baseline results demonstrate that over 50% of households experienced a change in their asset wealth status over time (Fig. 2a). The sex of the household head (an indication of an absent male head due to circular labour migration, polygamy, or AIDS-related mortality^{45,51}), being eligible for accessing institutional resources such as the old age grant (a primary source of stable income for rural populations⁵²), and baseline asset status have a significant influence on subsequent household asset wealth (Fig. 2b). “

[10] I was confused by the term “density” on the y-axis of Fig. 1b because network science uses “density” in a different sense in this figure. The authors should specify the term (“Degree” on the x-axis could confuse readers outside of network science).

[Reply 10] To avoid further confusion, we have revised the y-axis of Fig. 1 as Proportion, whereas the x-axis as number of ties.

[11] I did not understand what the network diagrams of the random walk represent in Fig 1c. Please specify them in the main text (or I would recommend removing them).

[Reply 11] This aspect was indeed confusing in our presentation. We have now included additional descriptive analyses to better contextualize our observed network data with topologies commonly observed in social network science. We follow this with an evaluation aimed at visualizing these local substructures as follows:

“Descriptively, this sparsely connected undirected network consists of 10,162 inter-household ties and follows a long-tailed degree distribution with an average degree of 1.72 (SD = 1.8) per household (Fig. 1b). This network has a global clustering coefficient of approximately 0.21 and an average path length of roughly 11.84 (SD = 2.98). Over 80% of households, approximately 8,453 in total, are connected through a shared

member. The largest connected component in this network consists of about 6,462 households (Fig. 1c). We compare this observed network with theoretical models within the broader landscape of network topologies, such as the random⁵¹, small-world³⁰, and scale-free²⁶ types (Supplementary Table 2). Compared to a random network, which has a lower clustering coefficient and a shorter average path length, our network shows a degree of clustering indicative of a more structured, possibly hierarchical arrangement. Although the average path length and local clustering are closer to that observed in small-world networks, the degree distribution does not align with the uniformity seen in such models, nor does it fit the hub-and-spoke configuration typical of scale-free networks. These observations suggest that our observed network may exhibit community structure with a quasi-small-world configuration, where a small share of social ties could play a role in enhancing overall connectivity and functionality³⁰.

To further explore the configurations of the constructed network, we delineate the local substructures of the largest connected component using random walks (Fig. 1c). In addition to the typical spanning-tree-like structure found in family lineage networks⁴⁵ (Random walk 3), these substructures uncover diverse embeddedness patterns that have not been widely reported in studies concerning the utility and economic functions of informal family support systems across lower-income settings¹³⁻¹⁷. We discuss the setting, data conceptualisation, and network construction method in greater detail in the Methods section and Supplementary Note 1.”

In closing, we wish to thank the Reviewer again for their time and for another round of careful reviews.

Reviewers' Comments:

Reviewer #1:

Remarks to the Author:

Thank you to the authors for their responses and the modifications they have made. Overall, I am satisfied with the authors' responses and the changes made to the manuscript. However, I still have reservations on several points, particularly regarding the network and econometric calculations. Nevertheless, I will refrain from further criticism as it is not my role as a reviewer to shape the methodology of the study according to my own preferences. The authors have demonstrated persistence in continuing with the analytical framework and techniques they began with. Moreover, since the manuscript has been cleared for another round of review despite previous criticisms, I believe this aspect is unlikely to change. On the other hand, it is possible that my observations are simply erroneous.

Therefore, I will provide only a few minor comments:

- Household wealth is based on a Principal Component about which we have no information.
- Random walks in Figure 1/c are still entirely redundant. They do not contribute to either the understanding of the network or the analysis.
- Figures 1/a and 1/b should be formatted similarly to Figures 4 and 5, as the current format appears to be in PowerPoint style.

Reviewer #3:

Remarks to the Author:

I am grateful again that the authors considered the second round of reviews, including mine, in their revision. The paper is now clear in all aspects of research questions, local contexts, data analyses, and their interpretations. This research will be a valuable contribution to social network theories and policy applications regarding the multi-layered network method and its findings on network interactions associated with wealth inequality. With this revision, I would recommend that this paper be published by Nature Communications.